# Episodic transport of discrete magma batches beneath Aso volcano

Jieming Niu [1,2✉] & Teh-Ru Alex Song[1,2]

Magma ascent, storage, and discharge in the trans-crustal magmatic system are keys to long-term volcanic output and short-term eruption dynamics. How a distinct magma batch transports from a deep reservoir(s) to a pre-eruptive storage pool with eruptible magma remains elusive. Here we show that repetitive very-long-period signals (VLPs) beneath the Aso volcano are preceded by a short-lived (~50–100 s), synchronous deformation event ~3 km apart from the VLP source. Source mechanism of a major volumetric component (~50–440 $m^3$ per event) and a minor low-angle normal-fault component, together with petrological evidence, suggests episodic transport of discrete magma batches from an over-pressured chamber roof to a pre-eruptive storage pool near the brittle-ductile transition regime. Magma ascent velocity, decompression rate, and cumulative magma output deduced from recurrent deformation events before recent 2014 and 2016 eruptions reconcile retrospective observations of the eruption style, tephra fallouts, and plume heights, promising real-time evaluation of upcoming eruptions.

[1] Seismological Laboratory, Department of Earth Sciences, University College London, WC1E 6BT London, United Kingdom. [2]These authors contributed equally: Jieming Niu, Teh-Ru Alex Song. ✉email: j.niu@ucl.ac.uk

Theoretical considerations have shown that the ascending of a magma batch is likely dictated by the density contrast between magma and surrounding rock[1], the excess pressure in the magma reservoir[1], magma viscosity[1], and the state of ambient stress[2]. More recent developments on magma transport and storage emphasize a trans-crustal magmatic plumbing system consisting of relatively long-lived crystal-rich mushes and shorter time-scale crystal-poor pools that were eventually tapped by eruptions[3,4]. On the other hand, after caldera-forming eruptions, the crystal-rich mush is largely consumed and post-caldera eruptions are of relatively small volumes but frequent, manifested by multiple cones and stratovolcanoes above a magma plumbing system consisting of multiple discrete chambers (or reservoirs)[5,6]. Nevertheless, the key process of distinct magma transport through these discrete chambers is not well known, even though it is intimately linked to magma ascent and discharge, controlling short-term eruption dynamics and long-term volcanic output.

Numerous petrological means have been utilized to interrogate magma ascent[7], including recent chemical diffusion-based geospeedometers[8,9]. However, these advancements rely on retroactive analysis of eruption products, which cannot be obtained in real-time during unrest or before eruptions. Furthermore, these estimates do not resolve the volume or duration of distinct magma ascent. Magma ascent inferred from the migration of seismicity can be difficult as magma can move aseismically if the walk rock is not close to failure. Furthermore, seismicity rarely follows simple upward movement and the progression of seismicity could indicate conduit formation or propagating of pressure through existing magma, rather than magma ascent[1,10,11].

Ground deformation detected from geodetic means (e.g., InSAR, GNSS, tiltmeter, strainmeter) provides invaluable insight into the average rate of magma supply[12–14]. Except few instances where syn-eruptive deformation of 0.1–10 microstrain over hours or less are reported[15,16], the documented pre-eruptive ground deformation is typically on the order of millimeter to centimeter, microstrain, or microradian over long durations of days, weeks, or months and they do not necessarily offer a sufficient resolution to assess the nature of a distinct magma ascent, potentially operating at a much shorter duration, higher frequency, or/and smaller volume.

Seismic resonances with periods of 0.2–200 s have been widely detected near shallow conduits or reservoirs (e.g., <2 km) in diverse volcanic systems[17,18]. They are often stationary and repetitive, responding to internal processes accompanying magma ascent or/and the build-up of conduit overpressure, including degassing, heat transfer and vaporization, change in the fluid pressure, or/and unsteady magma transport of a deeper origin[17,18]. The repetitive nature of shallow seismic resonance permits the use of stacking to tease out small deformation signals[19,20] that are otherwise undetected.

Beneath Aso volcano, an basalt-andesitic volcano in Japan[21–23], repetitive very long-period signals (VLP, ~15 s period) are known to occur in a fluid-filled crack-like conduit within a shallow hydrothermal aquifer[24–26], ~1 km beneath the active Naka-dake first crater[27–32] (Fig. 1a). Several VLP families exist in 2011–2016 and they share a largely common source location and geometry but differ somewhat in their resonance periods[32]. VLP families with a positive (negative) initial polarity are categorized as pressurization (depressurization) events in the shallow crack-like conduit[28,30,32]. The pressurization and depressurization VLPs have been attributed to vaporization and outgassing process, respectively[28,32]. Most importantly, VLP in Aso is known to occur during the quiescence and active period[28,30,32], offering a unique opportunity to methodically explore discrete magma ascents over the entire eruption cycle in 2011–2016, including changes in the crater-lake level, minor phreatic eruptions, the

incandescence followed by ash eruption, and Strombolian eruption in the late 2014[33]. The later stage of the eruption cycle involves wall collapse in early 2015, phreatomagmatic and explosive eruption in the late 2015 and late 2016, respectively[33].

Here we report new observations that manifest a causal relationship between VLP and synchronous deformation event within the deeper plumbing system beneath Aso volcano during the 2011–2016 eruption cycle. We analyze seismic and tilt waveforms recorded by the fundamental volcano observation network, or V-net[34], which is equipped with collocated surface broadband seismic sensors (Nanometrics 240, ~250 s natural period) and borehole tiltmeters (~1 nanoradian resolution) since early 2011. We systematically examine the presence and the stability of such synchronous signals and implement the matched-filter technique[32,35] against all VLPs during the 2011–2016 Aso eruption cycle[32]. Notably, the amplitude of the two VLPs prior to the 2016 eruption is at least 2 orders of magnitude larger than any VLPs we identified in 2011–2016[32] and highly correlated, we choose the first of the two VLPs (Event 1) as the reference event. We cross-correlate tilt and seismic waveform data of each VLP against the template waveforms in the ultra-long period band of 50–250 s. Such filtered waveforms show an amplitude–distance decay trend identical to that of the static waveforms in the near field (Supplementary Fig. 1), but with a high waveform coherency and signal-to-noise ratio, allowing us to effectively detect possible displacement or tilt offset in the near field (see Methods). Specifically, we gather detections with the same sign of cross-correlation coefficient in a given calendar time window to produce waveform stacks and compute bootstrap uncertainties[36].

## Results and discussions

**Discovery of the inflation/deflation event synchronizing with VLP.** We highlight tilt offsets accompanying the two anomalously large VLPs 2–3 minutes before the 8th Oct 2016 phreatomagmatic explosion[37] (Fig. 1b). In particular, the east–west tilt offsets accompanying the two VLPs reach ~58 nrad and ~88 nrad, respectively, at station N.ASHV. Similar observations of tilt and vertical displacement can be identified in the VLP catalog[32] (Supplementary Fig. 2). In general, the tilt (vertical displacement) offsets at station N.ASHV are either to the east (up) or to the west (down), but the amplitude is typically much smaller than those accompanying the two VLPs before the 2016 eruption. It is worth noting that these detected events are not necessarily associated with eruptions.

As shown in Fig. 1b, there are several striking differences between the VLP and the tilt offset at different stations and in the different components of the same station. First, the tilt offset at station N.ASHV is much stronger in the east–west component than that in the north–south component, whereas the VLP signal in the north–south component is much stronger than that in the east–west component. Secondly, the east–west tilt offset at station N.ASHV is much stronger than that at station N.ASIV, but the east–west component of VLP amplitude is much weaker than that at station N.ASIV. These observations strongly suggest that the source of the tilt offset is spatially separated from the VLP source, possibly closer to station N.ASHV than the source of VLP, which is near the active Naka-dake first crater[28–30,32] (see also Fig. 1a).

Here we present two global waveform stacks of the highest quality at station N.ASHV, highlighting observations during the volcanic unrest (e.g., dried-up of the crater-late, minor phreatic and ash eruptions and incandescent phenomena) in January 2011–August 2014 (Fig. 2a) and the intermittent Strombolian eruptions in October 2014–April 2015 (Fig. 2b), respectively. The timing of the inflation events in 2011–2014 does not necessarily correspond to isolated minor phreatic or ash eruptions. During the volcanic unrest, we observe an upward displacement offset

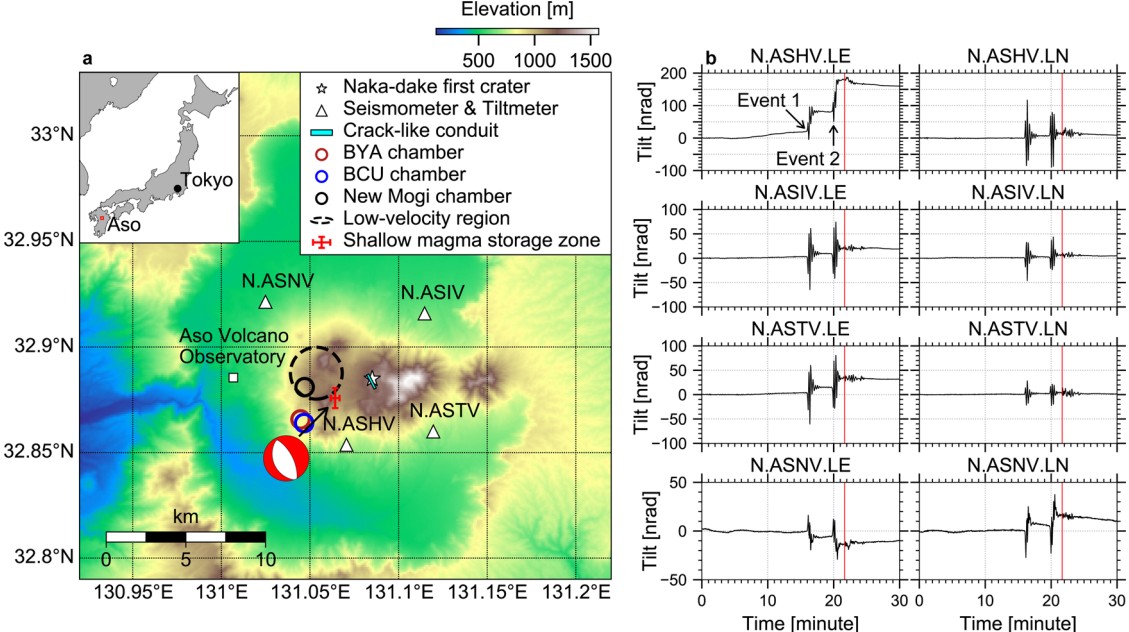

**Fig. 1 Aso volcano system and observations of synchronous VLP and tilt offset. a** The topographic relief near the Aso caldera is overlaid with collocated seismometer and tiltmeter (open triangles), VLP source in the shallow crack-like conduit[29] (thick bar), the geodetically inferred magma chamber[45,46] (open circle), and the low-velocity zone[47] (dashed circle). BCU and BYA chambers are inferred from levelling data in 1993–1994 and 1993–2004, respectively[45]. New Mogi chamber is inferred from GNSS (2004–2008) and levelling data (1998–2008)[46]. The star marks the active Naka-dake first crater. The upper inset displays Aso volcano located in Kyushu Island, southwest Japan. The source of the tilt offset is marked as a red cross, corresponding to a shallow magma storage zone (SMSZ) connected to the top of the chamber roof. The beach ball displays the deviatoric component of the source moment tensor. The depths of BCU, BYA, new-Mogi, low-velocity zone, and crack-like conduit can be referred to Fig. 3c. **b** Tilt waveforms at four V-net stations near the October 8, 2016 phreatomagmatic eruption. The waveforms are low-pass filtered at 0.05 Hz with a 4th-order casual Butterworth filter. The linear trend determined in the first 10 min is used to remove the background trend. "LE" and "LN" denote eastside and northside down tilts, respectively. "Event 1" and "Event 2" denote the tilt offset that occurred concurrently with two VLPs. The red line marks the timing of the 2016 eruption.

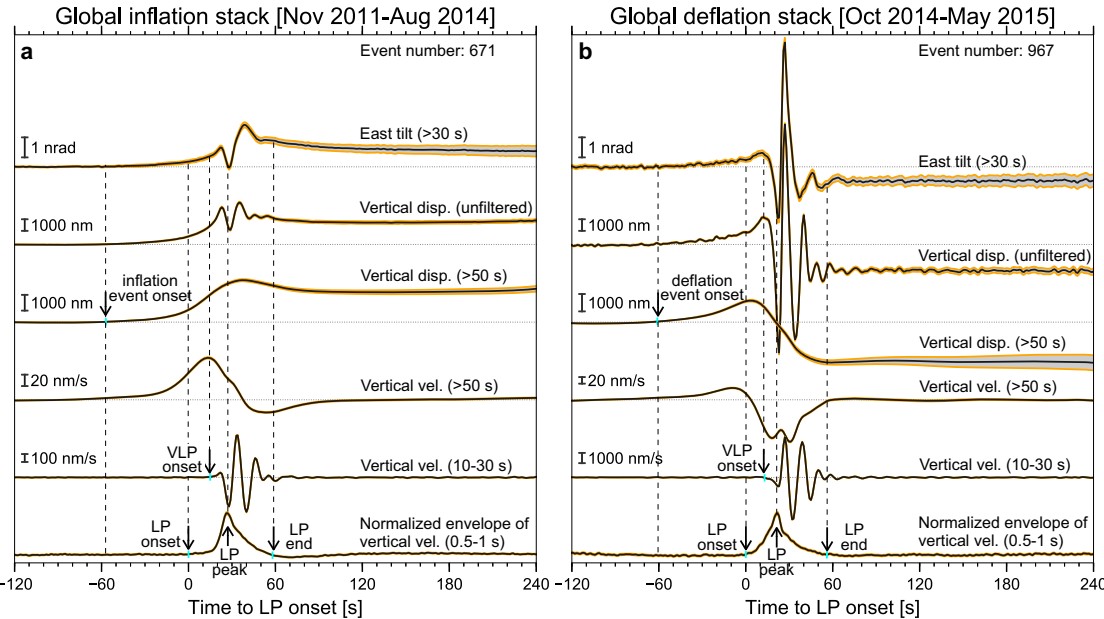

**Fig. 2 Observations of global waveform stacks against VLP and LP. a** Global waveform stacks from inflation events in the subset I. **b** Global waveform stacks from deflation events in the subset I. Broadband tilt, displacement, and velocity of global waveform stacks at station N.ASHV are presented against VLP velocity waveform stacks (10–30 s) and LP envelope stacks (0.5–1 s). The yellow strip marks the bootstrap uncertainties at the 99.7% confidence level. These waveforms and envelopes are aligned against the onset time of LP envelope stack. To preserve the phase of the long-period waveform (>50 s), a 2nd-order acasual Butterworth filter is implemented. We apply a 2nd-order causal Butterworth filter to preserve the timing of the VLP waveform stack (10–30 s) and LP envelope stack (0.5–1 s).

and a predominantly east-down tilt offset (Fig. 2a). During the intermittent Strombolian eruptions in 2014–2015, we observe a downward displacement offset and a predominantly west-down tilt offset (Fig. 2b). While the amplitude of the VLP differs by a factor of ~3 between the two episodes, the waveform offset is relatively constant, i.e., ~1 μm in the vertical displacement and ~1 nrad in the east–west tilt, respectively. Small bootstrap uncertainties of the global waveform stacks suggest the prevalence of the signal that is synchronous with VLP. Here we refer to the signal associated with an upward (downward) displacement offset as the inflation (deflation) event.

While it is well documented that VLP is often accompanied by long-period signal (LP, ~0.5–1 s period) located near the top of the VLP source region[17,28,38], Fig. 2 demonstrates that the inflation/deflation event occurs ~50 s earlier than LP or VLP, providing a causal source of internal triggering. While LP precedes VLP by ~10 s, the peak energy of LP envelope arrives ~10 s after the VLP onset. The inflation (deflation) event, VLP, and LP end at approximately the same time. Signals of inflation/deflation events can also be observed in other stations or high-quality monthly waveform stacks (Supplementary Fig. 3). Notably, the tilt offsets can remain relatively constant over 30 min (Supplementary Fig. 4). We also measure the displacement and tilt amplitude ratio between stations N.ASHV and N.ASIV in the ultra-long period band of 100–200 s (Supplementary Fig. 1) against all monthly stacks (Supplementary Figs. 5–10). The stability of these observational attributes indicates that repetitive inflation and deflation events share generally the same proximity with a relatively stationary source region.

**Source location and mechanism of the reference event**. We measure the vertical displacement, horizontal displacement (Supplementary Fig. 11, see Methods), and horizontal tilt offsets of the reference event (Event 1) and then invert the source location and mechanism following the Bayesian source inversion approach by Fukuda and Johnson[39] (Fig. 3a–c, see Methods). The tilt and displacement are modeled against isotropic, tensile-crack, dip-slip, and strike-slip source types in an elastic half-space[40]. Given the long-period nature of the signal (i.e., > 50 s), the specific choice of elastic constant does not introduce noticeable difference in the source location or/and mechanism. We find that the source is located at a depth of ~3 km below sea level, ~2 km west of the Naka-dake first crater (Fig. 1a, Fig. 3a–c, see also Supplementary Table 2). The reference event has a moment magnitude $M_w = 3.3$ with a predominantly isotropic component (~80%) and a minor dip-slip (normal-fault) component of ~20% (Fig. 1a, Supplementary Fig. 12, Supplementary Table 2), whereas

the tensile-crack component is generally very small (i.e., <<10%). The corresponding volume change from the isotropic component and the tensile-crack component is ~25,800 ± 6070 m³ (see Methods and Supplementary Table 2).

**The nature of inflation/deflation event in Aso magma plumbing system**. As shown in Fig. 2, the inflation or deflation event consistently occurs ~50 s before LP and VLP, providing a causal link to the triggering processes of LP and VLP beneath Aso volcano. We speculate that the rising stress during the initial stage of the inflation and deflation event may eventually exceed the tensile strength of the conduit plug above the shallow crack-like conduit, resulting in fractures and rapid leakage of gas or fluid-rock mixture, generating LP and causing the aquifer to shrink and resonates, which results in VLP[28].

Figure 4a provides a sketch to summarize the renewed magma-plumbing system beneath Aso volcano. After the most recent Aso-4 caldera forming eruption[41,42], a crystal-rich mush reservoir likely remains at a deeper depth (i.e., >10 km) during post-caldera eruptions[42,43], supplying volatile-rich basaltic magma toward the magma chamber. The volatile-rich basaltic magma is thought to mix with volatile-poor silicic magma in the chamber[6,43,44], and the mixed magma is stored at a shallower storage zone above the chamber before the eruptions.

The source of the inflation/deflation event is probably located near the inferred magma chamber[45–47] (Fig. 1a), but at a shallower depth similar to the previously detected reflection void[48], corresponding to a shallow magma storage zone (SMSZ)[43] (Fig. 3c). The SMSZ connects the source of VLP in the shallow crack-like conduit from above and the magma chamber from below (Fig. 4a), corresponding to a zone of relatively high electrical conductivity (i.e., tens of Ω·m)[25,49]. The depth of the SMSZ is very consistent with the highest gas saturation pressure of melt inclusions in scoria (~80–118 MPa, or 2–3.5 km below sea level)[21], supporting the hypothesis that the magma (and gas) are likely transferred from the magma chamber and temporarily stalled in the SMSZ (i.e., a pool of mixed magma), which serves as a preparation zone for the storage of mixed magma before upcoming eruptions.

The recurrences of these deformation events are potentially regulated by the brittle–ductile transition rheology under high temperature (~400 °C)[50]. High strain-rate during the surge of magma flux from below may also promote brittle failure[51]. We suggest that inflation/deflation event manifests short-lived (i.e., 50–100 s), episodic transport of a discrete magma batch from the roof of the magma chamber to the SMSZ. The transport of

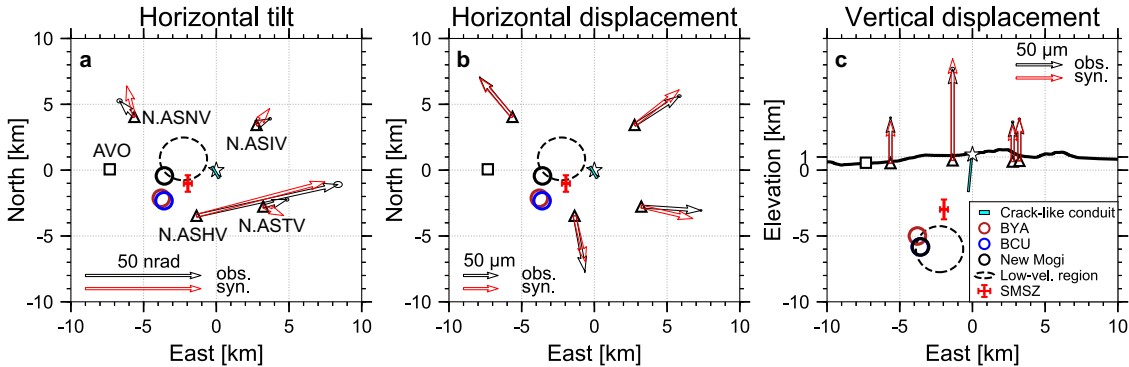

**Fig. 3 Source inversion result. a–c** Comparisons between observations (black arrows) and synthetics (red arrows) predicted from the inverted source location and mechanism. Red cross marks the inverted source location with uncertainties of 99.7% confidence interval and it corresponds to the shallow magma storage zone (SMSZ). BYA, BCU, New Mogi: magma chambers inferred from geodetic studies[45,46] (see also Fig. 1a); AVO: Aso Volcano Observatory.

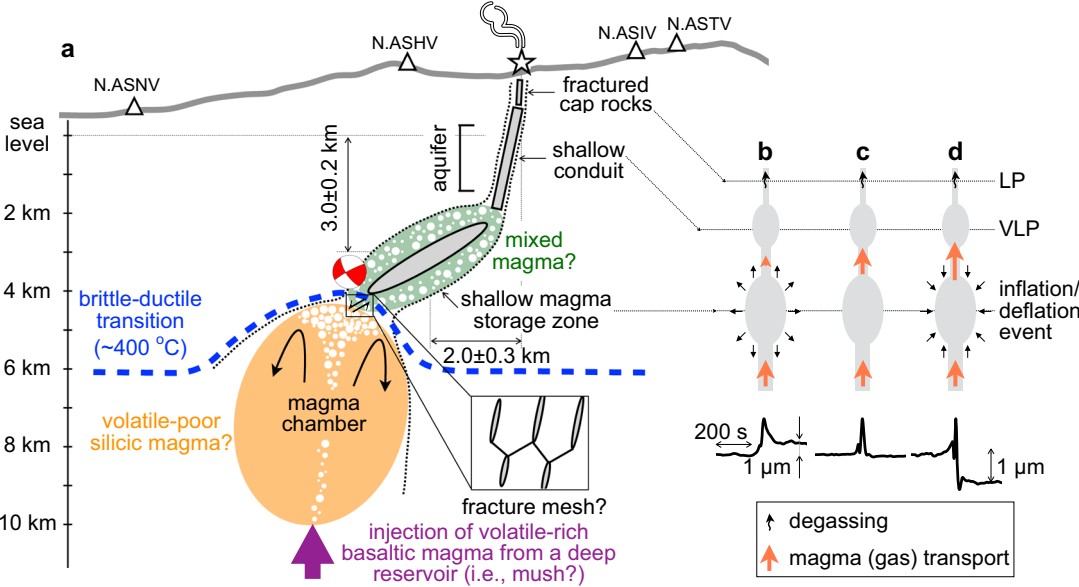

**Fig. 4 A conceptual framework of magma/gas transport in the plumbing system beneath Aso volcano. a** An east–west cross section showing a conceptual diagram of magma/gas transport system beneath Aso volcano. The inferred source is located between the VLP source in the shallow crack-like conduit from above and the magma chamber from below. LP source is shown directly above the shallow crack-like conduit. The beachball displays the deviatoric components of the estimated focal mechanism, looking from the south. Blue dashed line indicates the brittle-ductile transition following the 400 °C isotherm. Black curvy arrows indicate magma mixing and white voids indicate exsolved gas. The inset displays fracture network near the low-angle west-dipping normal fault. **b–d** The simplified flow conditions where the SMSZ suffers inflation, null volume change, or deflation, respectively. Waveform stacks indicative of these three conditions are shown directly below.

discrete magma batches may include volatiles that may readily exist in the top of the chamber.

Gases near the top of the magma chamber may facilitate episodic brittle failure due to increasing magma buoyancy. The presence of unfavorably oriented low-angle normal fault near the roof of magma chamber also supports sustained overpressure[52,53]. When the buoyancy of gassed magma and overpressure exceed the tensile strength near the chamber roof, the transport of a discrete batch of magma (and gas) may potentially be accommodated by choked flows along a nozzle or conduit[54].

Alternatively, the low-angle dipping normal fault transects the over-pressured chamber roof can behave as a fluid-pressure activated valve, becoming permeable channels post-failure. This scenario is consistent with the observations that the initial motion of the inflation/deflation event does not differ, consistently moving in the same fashion (Fig. 2). While the fault-valve behavior typically occurs near the brittle–ductile transition in the bottom of seismogenic zone under lithostatic pore-pressure, it is conceivable that such processes can also take place under the brittle–ductile transition regime in an over-pressured magmatic-hydrothermal system[55].

We hypothesize that fault-valve actions may facilitate intermittent discharge of magma (and gas) from the over-pressured chamber roof to the SMSZ, perhaps through a network of small fractures[52,53] (Fig. 4a). The deformation event ends as the system returns to the ductile regime as a result of the shear-stress release, decreasing magma supply (i.e., lower strain rate) or/and lower fluid pressure. We note that the duration of each deformation event (on the order of 10–100 s) is much longer than what is expected for crustal earthquakes of similar size (e.g., <<1 s for a Mw 3 earthquake). However, such a slow deformation, to some extent, is analogous to slow-slip events observed worldwide in the subduction zone interface or in the deep crust[56], where the rheological condition and fluid pressure are likely similar to that near the chamber roof.

We suggest that, when the inflow (outflow) is higher than the outflow (inflow), there is a net volume increase (decrease) in the SMSZ and we observe an inflation (deflation) event (Fig. 4b, d). When the inflow is approximately equal to the outflow, there is negligible net volume change in the SMSZ and we observe a negligible offset in tilt or displacement (Fig. 4c). As shown in Fig. 5a, we observe predominantly inflation events during crater dried-up and rising crater bottom (wall) temperature and $SO_2$ emissions in July and August 2014 (Fig. 5b, c).

Right after the ash eruption at the end of August 2014, we observe events with a minimal net offset in September 2014. During the intermittent Strombolian eruptions (e.g., November 2014, March 2015), we observe predominantly deflation events. After the pyroclastic cone collapsed in early May 2015, we again observe inflation events in May and June 2015. While the deflation events dominate in late 2015 and most of 2016, two anomalously large inflation events occur just minutes before the 2016 phreatomagmatic eruption (Fig. 1b).

These systematic observations point to a robust and intimate link between surface volcanic activities and the state of the SMSZ. In particular, GNSS displacement from JMA show notable inflation of the magma chamber in July 2014, May 2015, and July 2016, suggesting magma ascent from a deep reservoir (>~10 km) toward the magma chamber[33]. Furthermore, we observe a strong correlation between a substantial increase in the number of inflation events and magma transport toward the SMSZ (Fig. 5a), rising crater bottom (wall) temperature (Fig. 5b) and $SO_2$ emission (Fig. 5c). Notably, the rising of inflation events apparently leads up to the increase of crater bottom temperature, which precedes the increase of $SO_2$ emission.

**Pre-eruptive volume change in SMSZ and upcoming magmatic eruptions**. Our new observations suggest that the upward transport of magma/gas from the magma chamber toward the surface in the 2011–2016 Aso eruption cycle is a stepwise process

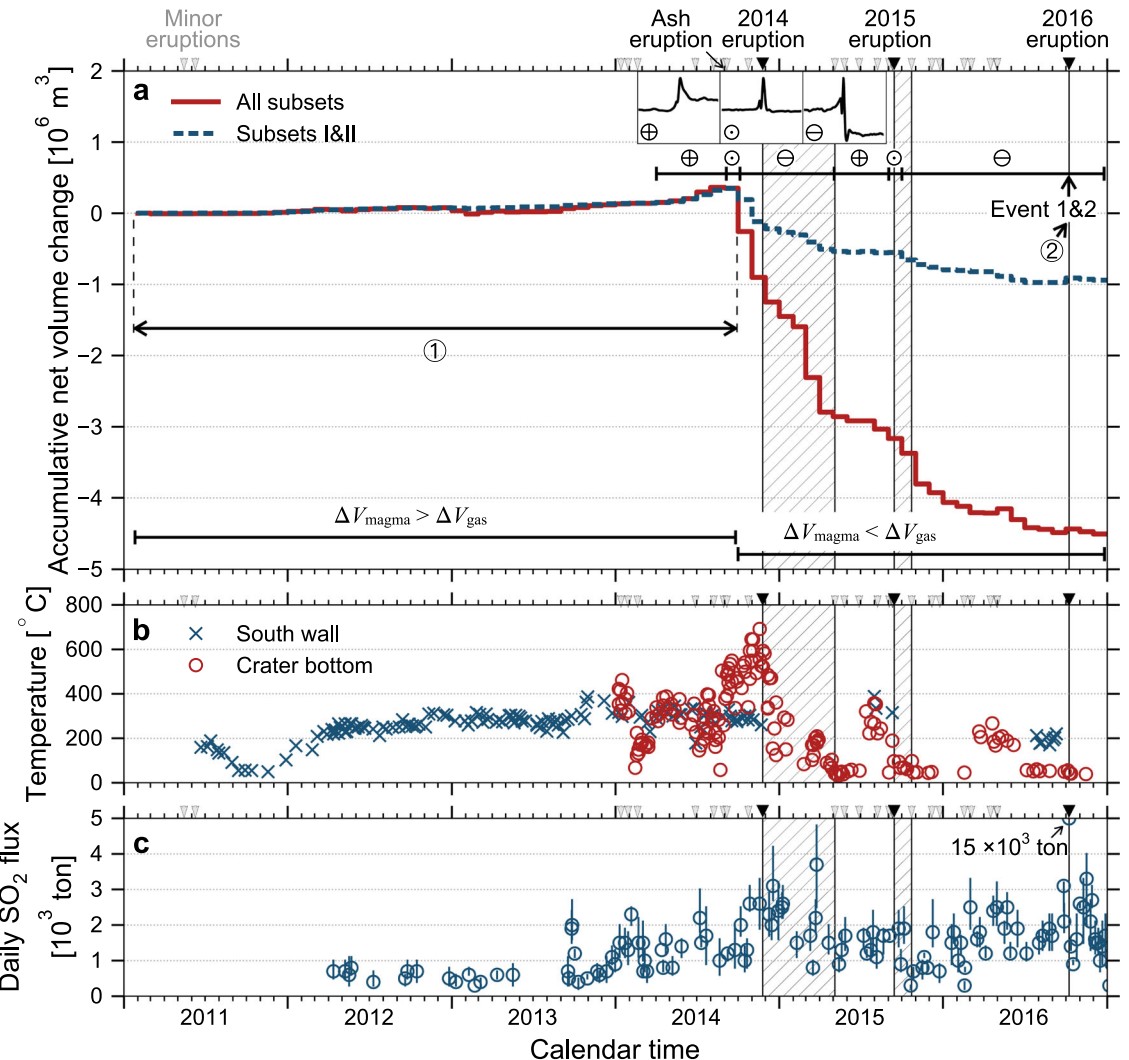

**Fig. 5 Tracking the net volume change in the SMSZ, crater bottom (wall) temperature, and SO₂ emission during the 2011–2016 eruption cycle. a** The accumulative net volume changes in the SMSZ. Dark inverse triangles mark the onset of the 2014 Strombolian eruption, the 2015 and 2016 phreatomagmatic eruptions, whereas hatched areas mark their eruption episodes. Grey inverse triangles indicate minor eruptions. The volume changes from the high-quality subsets I+II are shown in blue dotted line. Enclosed plus, minus, and dot mark the episodes associated with observations of prominent inflation event, deflation event, and event of negligible offset, respectively (see waveform stacks in the insets). Enclosed 1 and 2 mark the time windows where the mass changes in the SMSZ before the 2014 and 2016 eruptions are calculated, respectively (see also Fig. 6a). **b** Crater-bottom temperature measured from the Naka-dake first crater (red circle), provided by JMA[33] and Cigolini et al.[79]; South wall temperature (blue circle) is provided by JMA[33]. **c** SO₂ emission from the campaign ground-based sensor, provided by JMA[33].

in an episodic fashion. While these deformation events are indicative of magma transport, they are not necessarily associated with surface eruptions, which are not only dictated by the over-pressure (i.e., magma supply), but also the strength/permeability of the conduit plug[39].

Since the source is repetitive, the volume change associated with an individual inflation/deflation event can be obtained by scaling the tilt amplitude of each event against that of the reference event (see Methods). For example, the east–west tilt of the reference event at station N.ASHV is ~58 nrad, corresponding to a volume increase of ~25,800 ± 6070 m³. The typical east–west tilt in the global stacks is only about ~1 nrad (Fig. 2), which translates to a volume change of ~440 ± 105 m³ per event, or the size of a typical swimming pool.

We systematically measure the tilt amplitude of the monthly stacks (Supplementary Fig. 13, see Methods) and estimate the volume change per event. In the subset of the highest (lowest) signal-to-noise ratio, the volume change per event is

~440 (50) m³ (Supplementary Fig. 14). We calculate the monthly volume change by multiplying the volume change per event against the event number (Supplementary Fig. 14) and the cumulative volume change over 2011–2016 is illustrated in Fig. 5a.

For a magma density of 2500 kg/m³, we convert the net volume change $\Delta V$ shown in Fig. 5a to obtain the net mass change in the SMSZ. The effect of magma compressibility on $\Delta V$ has been calibrated against the estimated volatile contents in melt inclusions of scoria[21] and it is relative minimum, i.e., a factor of 1.1–2.3 (See Methods). There is a good agreement between the total mass change in the SMSZ before the 2014 eruption and the mass of tephra fallout[57] (Fig. 6a).

While the net volume change $\Delta V$ in the SMSZ can be due to magma ($\Delta V_{magma}$) or/and gas ($\Delta V_{gas}$), i.e., $\Delta V = \Delta V_{magma} + \Delta V_{gas}$, our observation shown in Fig. 6a is consistent with the scenario where $\Delta V \sim \Delta V_{magma}$ (and $\Delta V_{magma} \gg \Delta V_{gas}$) before the 2014 eruption. However, the net volume change in the SMSZ

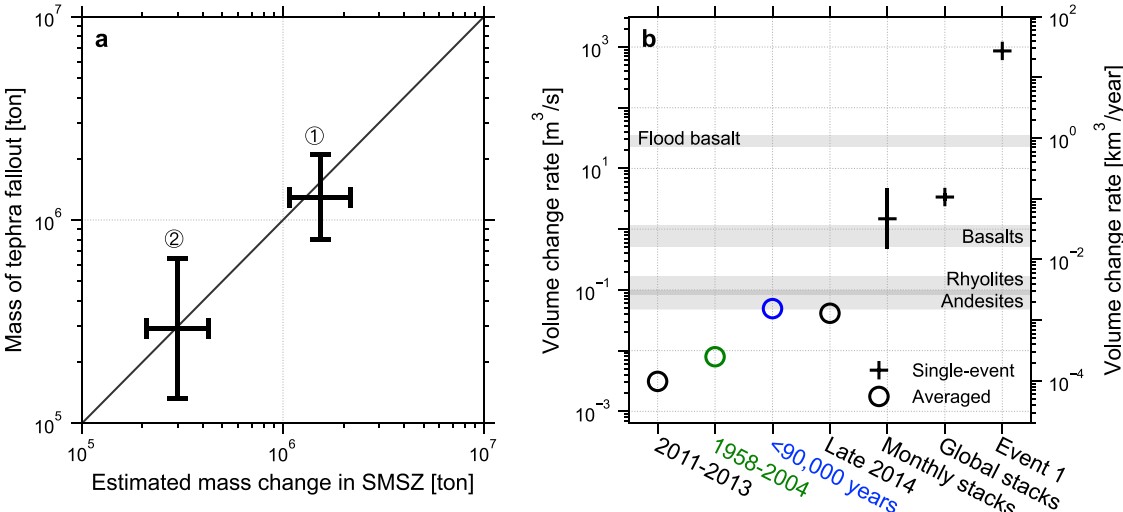

**Fig. 6 Pre-eruptive mass change in the SMSZ versus retrospective tephra fallout mass and contrast of volume change rates across multiple time scales in Aso volcano. a** The observed volcanic output of the 2014 Strombolian and 2016 phreatomagmatic eruptions against the estimated mass change inside the SMSZ before the eruptions. Enclosed 1 and 2 correspond to the time windows defined in Fig. 5a. Horizontal bar shows the uncertainties of estimated mass change due to the magma compressibility (see text and Methods). Solid line shows the 1:1 reference line. **b** Comparing the single-event volume change rates (black cross) against the averaged volume change rates over 2011–2013 and late 2014 (black circles) (Fig. 5a), the geodetic data over 1958–2004[45] (green circle) and the geological analysis over post-caldera eruptions[23] (blue circle). Global averaged volcanic output rates from White et al.[80] are shown in the gray shaded regions.

during/after the 2014 Strombolian eruption episode is 6–8 times larger than that before the eruption (see also Fig. 5a). It suggests that the estimated net volume change during the eruption is not only associated with magma, but also, to a much larger extent, contributed by gas ($\Delta V \sim \Delta V_{magma} + \Delta V_{gas}$ and $\Delta V_{gas} > \Delta V_{magma}$).

Conceivably, the tilt offset in some of the deformation events may decay beyond the time scale of our analysis (i.e., $> \sim 1$ h). Consequently, we overestimated the volume change associated with magma flux in the SMSZ. Such deflation events are likely of a small magnitude with a lower signal-to-noise ratio. This scenario is supported by the net volume change from events with a high signal-to-noise ratio (i.e., subset I+II) (Fig. 5a), i.e., the estimated net volume change before the eruption remains the same, and the net volume change in the SMSZ during/after the eruption is much lower and more compatible with the net volume change before the eruption.

As shown in Fig. 6a, the mass change of the two anomalously large inflation events ~2–3 min before the 2016 phreatomagmatic explosion is also readily comparable to the mass of juvenile magma from the ash fallout and pyroclastic current deposit (PDC)[58]. Nevertheless, if the SMSZ buffers the upward magma/gas from the magma chamber, our observation is consistent with the scenario where the cumulative mass change in the SMSZ is closely linked to the size of upcoming magmatic eruptions.

**Single-event volume change (rate) vs. long-term averaged volume change rate**. Considering a typical source duration of ~100 s (Fig. 2), the single-event volume change rate estimated from the global stacks or the monthly stacks is ~0.5–4.4 m³/s or ~0.03–0.13 km³/year, which is ~2 orders of magnitude higher than the averaged volume change rate in 2011–2013 (i.e., ~$10^{-4}$ km³/year) and in late 2014 (~$1.3 \times 10^{-3}$ km³/year), the geodetic deflation rate of the magma chamber at a greater depth (i.e., ~$2.5 \times 10^{-4}$ km³/year over several decades)[45] and Aso post-caldera output rates (i.e., ~0.001 km³/year)[22,23,44] (Fig. 6b).

We hypothesize that the volume change rate associated with an individual inflation/deflation event marks the instantaneous magma transport rate of a distinct magma batch, whereas the

recurrence interval modulates the averaged transport rate, reconciling the magma transport over multiple timescales. This hypothesis reconciles similar disparities noted in basaltic volcanic systems such as Kilauea[54] and silicic volcanic systems such as Mount St. Helens[59]. Magma composition, viscosity, and tectonic settings can potentially dictate the recurrences of such episodic deformations.

**Contrasting magma ascent velocity and mass flow rate near the SMSZ**. Using a simplified 1-D conduit flow model that balances the conduit radius against the overpressure modulated by the magma flow[60], the single-event volume change rate allows us to infer the magma ascent velocity and decompression rate near the SMSZ prior to the 2014 Strombolian eruption and the 2016 phreatomagmatic explosion (see Methods). For a low-viscosity magma (i.e., $10^3$ Pa·s), a density difference between magma and wall rock of ~100 kg/m³, a lithological gradient of 0.025 MPa/m, and a volume change rate of ~0.5–4.4 m³/s, we estimate the ascent velocity near the SMSZ of ~0.14–0.41 m/s and the decompression rate of ~0.004–0.011 MPa/s, which is comparable to syn-eruptive decompression rates estimated by melt embayment and water-in-olivine studies in basaltic eruptions[8,61] and theoretical estimates for the Strombolian eruption[62] (Fig. 7). Given a magma density of 2500 kg/m³, we can convert the single-event volume change rate (~0.5–4.4 m³/s) to the mass flow rate of ~1250–11,000 kg/s (Fig. 7).

In contrast, the two anomalously large inflation events ~2–3 minutes before the 2016 phreatomagmatic explosion corresponds to a much higher volume change rate of ~860–1300 m³/s over a 30 s duration. The estimated mass flow rate is ~$2.2–3.3 \times 10^6$ kg/s (Fig. 7) and the estimated ascent velocity and decompression rate near the SMSZ are also much higher at ~7 m/s and ~0.2 MPa/s, respectively, indicating a much larger overpressure.

While VLP is not always associated with surface eruptions, some VLPs do occur before the intermittent Strombolian eruptions in 2014–2015[63]. If the Strombolian eruption is triggered by gas burst ascending from the VLP triggering depth of

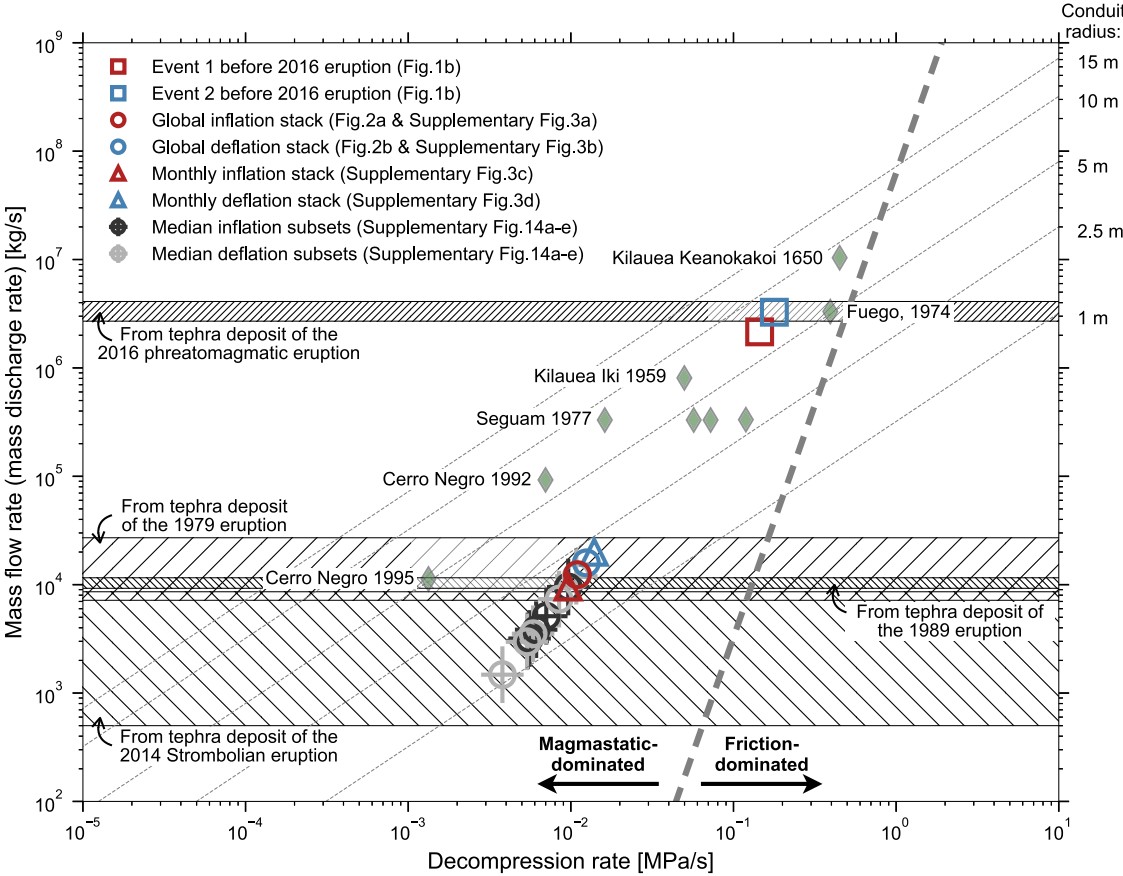

**Fig. 7 Comparisons of the mass flow rate and decompression rate from selected basaltic volcanoes and Aso: petrology and seismology.** Open circles and triangles mark the volume estimates from waveform stacks of inflation and deflation events before the 2014 and 2016 eruptions (see details in the main text). Target crosses mark the median volume estimates from the five subsets (Supplementary Fig. 14a–e). Green diamonds mark previous compilations in selected basaltic volcanoes[8], where the decompression rates are estimated from water diffusion in olivine or embayment studies and the mass discharge rates are estimated from volcanic outputs or plume heights, respectively. Contour lines display theoretical estimate of decompression rate and mass flow rate against fixed conduit radius (see equation 4 in Barth et al.[8]). Hatched areas show the mass discharge rate of the 1979, 1989, 2014, and 2016 eruptions in Aso volcano (see details in the main text). Dashed line indicates the boundary of magmastatic-dominated and friction-dominated decompression after Barth et al.[8].

~300–500 m[63], a 2–3 s delay time between the Strombolian eruption and the onset of VLP requires the gas ascent velocity on the order of ~$10^2$ m/s, consistent with annular flow regime. Such a triggering process is presumably in line with a slow magma ascent in the SMSZ prior to the Strombolian eruption.

On the other hand, the time delay between the VLP (i.e., Event 2) and the onset of 2016 phreatomagmatic explosion is much longer of ~120 s. While it is possible that rising gas slugs of ~2.5–4.2 m/s can lead to the 2016 phreatomagmatic eruption, the relatively fast magma ascent (i.e., ~7 m/s) near the SMSZ perhaps can easily reconcile such a time delay, if the magma ascends toward a wider shallow crack-like conduit, leading to such an explosive eruption.

Since the Strombolian eruptions in Aso are intermittent, the averaged mass discharge rate estimated by tephra deposits of the 2014 Strombolian eruption (i.e., ~25–430 kg/s)[57] is likely lower than the instantaneous mass discharge rate. Assuming only small percentages (i.e., 5%) of the entire eruption duration[22], the magma discharge rate of the 2014 eruption is ~500–8600 kg/s, and comparable to the mass discharge rate of the previous eruptions in 1979 and 1989[22]. The estimated single-event mass flow rate in the SMSZ prior to the Strombolian eruption (i.e., 1250–11,000 kg/s) are evidently very compatible with these mass discharge rates (Fig. 7).

Considering the tephra mass and the duration of the 2016 phreatomagmatic explosion of 160–260 s, the total mass discharge rate is 2.7–4.1 × $10^6$ kg/s[64], comparable to our inferred mass flow rate in the SMSZ (i.e., 2.2–3.3 × $10^6$ kg/s) (Fig. 7). Applying the empirical scaling between the mass discharge rate and the plume height[65], the mass flow rate in the SMSZ before the 2016 eruption projects a plume height of ~10 km, consistent with observations[33].

Comparing the mass discharge rate between basaltic and basalt-andesitic volcanoes can be useful in elucidating the regime where the mass discharge rate can be approximated by the mass flow rate (Fig. 7). Presumably, the fragmentation level is likely shallow with respect to the conduit length and the mass flow rate is simply proportional to the conduit radius, independent of magma viscosity[66] (e.g., basaltic vs. andesitic). As shown in Fig. 7, for a given decompression rate, the mass discharge rates (or mass flow rate) of basaltic and basalt-andesitic volcanoes are directly proportional to the conduit radius. While more data may be needed to validate if this observation holds, such an equality permits the inference of the mass discharge rate prior to the upcoming eruption.

**Perspective and outlook.** We illustrated that repetitive VLPs in Aso volcano provide a pathway to unravel magma flows deep in the magma plumbing system. Episodic transport of discrete

magma batch between multiple chambers or storage zones could be accommodated by short-lived (i.e., 100 s) episodic deformation event under brittle-ductile transition regime in an over-pressured magmatic-hydrothermal system. While persistent VLPs in Aso have been documented for almost a century, modern geodetic and broadband seismic sensor were unavailable during historical eruptions and observations of the inflation or deflation event cannot be made. However, establishing the amplitude scaling between VLP and the inflation (or deflation) event in recent eruptions potentially permit the estimate of pre-eruptive volume change (or volume change rate) in the SMSZ that is otherwise inaccessible for historical eruptions.

As the mass discharge rate and the magma ascent velocity or decompression rate can only be obtained retroactively after the eruptions[8–10,67], new seismic observations reported here underscore the possibility that, through real-time signal detection/processing and source analysis in volcanic systems where repetitive shallow seismic resonances (e.g., VLP or LP) are frequently detected[17,18] (e.g., Etna, Taupo, Asama, Kusatsu-Shirane), the volume change and volume change rate (or mass flow rate) of episodic deformation similar to observations in Aso could facilitate in-situ characterization of magma ascent and transport before upcoming eruptions. Such efforts can be coupled with high-resolution analysis of deep low-frequency earthquakes (DLFEs)[17,18,68,69] to unravel deep magma transport. Improved knowledge of external loading/unloading conditions[67], continuous monitoring of degassing rate[70], and the permeability/rheology of shallow conduit[32,67] can facilitate diagnosing short-term eruption potential.

## Methods

### Systematic detection using the matched filter (the inflation/deflation event).
A robust observation of the tilt or displacement offset shown in Fig. 1b and Supplementary Fig. 1 requires a high signal-to-noise ratio that is not routinely achievable. Instead, we measure the energy in the ultra-long-period band and use it as a proxy for the static offset[31]. Here we use the matched filtered technique[32,35] to systematically examine the presence and the stability of such synchronous signals against the VLP catalog during the 2011–2016 Aso eruption cycle[32].

The first of the two VLPs before the 2016 eruption (Event 1 in Fig. 1b) is selected as the reference template event. The waveforms at two long-running stations, N.ASHV and N.ASIV, are cut in a 10-minute window centered at the VLP arrival time and filtered in the period band of 50–250 s using a 3rd-order causal Butterworth filter to obtain the template waveforms. The waveforms of each VLP are processed in the same way as the reference event before the matched filter is applied to cross-correlate waveform data against the template waveforms. To avoid the coupling between tilt and translation in the horizontal components[71], we only concern vertical-component seismic waveform data and borehole tilt waveform data in the signal detection. Depending on the cross-correlation coefficient (CC) and signal-to-noise ratio (SNR) between waveform data and the template waveforms (Supplementary Table 1), the detection against the entire VLP catalog of more than 200,000 events is divided into five subsets (I–V) (Supplementary Table 2).

### Waveform stacking.
After removing the mean and trend of raw seismic waveforms, we align and stack the waveforms against the VLP occurrence time to produce the waveform stacks, which greatly enhances the signal and eliminates incoherent noise. To minimize the bias introduced by arbitrary spikes as well as the energy from large earthquakes, we normalize raw velocity waveforms of the single event against their vertical peak-to-peak velocity in the period band of 10–30 s at station N.ASHV before stacking. After stacking the normalized waveforms, the waveform stacks are multiplied by the median vertical peak-to-peak velocity in the period band of 10–30 s to recover the amplitude of the waveform stack in each month. The bootstrap technique is used to quantify the uncertainties of the waveform stacks[36]. After deconvolving the instrument response up to 1000 s, we remove the background mean in the first 150 s of the velocity waveform stacks, perform a causal low-pass filter, and integrate the velocity waveform stacks to obtain displacement waveform stacks.

Regarding tilt waveform stacks, we first differentiate the raw tilt waveform to obtain the tilt-rate waveform and apply the same data processing procedure as done against velocity waveforms. This stacking procedure not only reduces noise but also helps remove unwanted bias associated with tides, pressure, temperature, or rainfall, which are nonetheless trivial in the data window of 10 min.

### Estimate the static displacement offset from broadband seismograms.
As the seismic recording is also sensitive to sensor tilt, it is often difficult to recover true translational signal[72]. This tilt-related signal is often exhibited as a non-linear drift in the original displacement seismogram[72]. Due to the projection of gravitational force into the horizontal axis, this phenomenon is particularly severe in the horizontal components. Here we estimate and remove the nonlinear drift in the displacement seismogram by least-square fitting the portion of seismogram without major signals with a polynomial function[73].

The pre-event window is set to 20 s and the total duration of seismograms is 120 s. A 4th-order polynomial function is used as suggested by Zhu[73]. After removing the non-linear drift (Supplementary Fig. 11), the displacement offset in the individual component is calculated by differing the average displacements in the post-event (mean:$d^1$, standard deviation: $\sigma^1$) and pre-event (mean: $d^0$, standard deviation:$\sigma^0$) windows as $d = d^1 - d^0$, with uncertainty, $\sigma = \sqrt{(\sigma^0)^2 + (\sigma^1)^2}$.

The static displacement from this approach may be affected by the selection of polynomial order and time window[74]. In our case, the determination of the time window of the major signal of the reference event is relatively straightforward. As presented in Supplementary Table 2, the selection of polynomial order does not significantly change the estimations of the point-source location and source geometry.

### Source properties of the inflation/deflation event.
We followed the mixed linear–nonlinear source inversion scheme in a Bayesian framework proposed by Fukuda and Johnson[39] to obtain the location and mechanism of the reference event (Event 1 in Fig. 1b). This mixed inversion method efficiently handles non-linear inversion problems when some of the parameters are linearly related to the observations. The offsets of the vertical displacement, horizontal displacement, and horizontal tilt are modeled with Okada's analytic deformation model for a point source[40].

Considering a mixed linear-non-linear inversion problem[39] associated with a point dislocation/force source proposed by Okada[40]: $d=G(m)s+\epsilon$ where $G(m)$ is the kernel matrix relating non-linear model parameters, $m = [x_s, y_s, z_s, \theta, \phi]^T$ (3-D Cartesian location of a point source with respect to the Naka-dake first crater, strike and dip angles of the fault plane and crack plane), and linear model parameters, $s_1 = [m_{ss}, m_{ds}, m_{tc}, m_{ex}]^T$ (the scalar moments of strike-slip, dip-slip, tensile crack and explosion) or $s_2 = [m_{ss}, m_{ds}, m_{tc}, m_{ex}, f_E, f_N, f_U]^T$ (the scalar moments of strike-slip, dip-slip, tensile crack and explosion, the magnitudes of forces towards east, north, and up directions), to the observation vector, $d = [d_1, d_2, d_3]^T$ (vertical displacement, horizontal displacement, and horizontal tilt), with the error vector, $\epsilon = [\epsilon_1, \epsilon_2, \epsilon_3]^T$ where $\epsilon_i \sim N(0, D_i)$, following a normal distribution. $J^{th}$ minimum norm constraints (or Tikhonov regularization) are applied on the linear parameters, $||R_j s||^2$, where $J = 2$ if single-force is considered. Otherwise, $J = 1$.

An optimal solution can be found by minimizing an objective function $\Phi(m, s) = \sum_{i=1}^{I} \frac{1}{\sigma_i^2}[d_i - G_i(m)s]^T D_i^{-1}[d_i - G_i(m)s] + \sum_{j=1}^{J} \frac{1}{\beta_j^2} ||R_j s||^2$ where $\sigma_i^2$ adjust relative weights of the multiple data sets in fitting the data; $\beta_j^2$ adjust relative weights of regularization prior on $s$. If $J = 1$, $R_1 = I$. If $J = 2$, $R_1 = \begin{pmatrix} I_{4 \times 4} & 0 \\ 0 & 0 \end{pmatrix}$ and $R_2 = \begin{pmatrix} 0 & 0 \\ 0 & I_{3 \times 3} \end{pmatrix}$. Following Fukuda and Johnson[39], if assuming that the prior probability density functions of the linear, non-linear and hyperparameters are uniform, the posterior probability density function of the parameters given the data, $p(m, s, \sigma, \beta | d)$, can be given according to Bayes' theorem as below:

$$-\ln p(m, s, \sigma, \beta | d) \propto \sum_{i=1}^{I} 2N_i \ln \sigma_i + \sum_{j=1}^{J} 2M_j \ln \beta_j + \Phi(m, s^*) + \ln \left| \sum_{i=1}^{I} \frac{1}{\sigma_i^2} G_i^T(m) D_i^{-1} G_i(m) + \sum_{j=1}^{J} \frac{1}{\beta_j^2} R_j^T R_j \right|$$

where $s^*$ is the weighted damped least-square solution of the linear parameter given the nonlinear and hyperparameters, $m$, $\sigma$ and $\beta$; $N_i$ is the number of samples in $i^{th}$ data set; $M_j$ is the number of parameters efficiently regulated by $j^{th}$ prior constraint. If $J = 1$, $M_1 = 4$. If $J = 2$, $M_1 = 4$ and $M_2 = 3$.

To tackle the weights between displacement and tilt, we first normalize the kernel matrix $G(m)$, the observation vector $d$ and the covariance matrix $D$ by the absolute of observation vector as $K = \begin{pmatrix} 1/|d_1| & \cdots & 0 \\ \vdots & \ddots & \vdots \\ 0 & \cdots & 1/|d_N| \end{pmatrix}$, $G'(m) = KG(m)$, $d' = Kd$, and $D' = KDK^T$. Then, by utilizing Cholesky factorization, we equalize the weight difference among the observations as $D' = L^T L$, $d'' = Ld'$, $G''(m) = LG'(m)$. Then, if let $\sigma_i^2 = 1$, we can rewrite the objective function as $\Phi(m, s) = (y - As)^T (y - As)$ where $y = [d'', 0]^T$ and $A = \left[ G''(m), \frac{R_0}{\beta_0}, \cdots, \frac{R_j}{\beta_j} \right]^T$. In this case, the posterior probability density function of the parameters can be rewritten as $-\ln p(m, s, \beta | d) \propto \sum_{j=1}^{J} 2M_j \ln \beta_j + \Phi(m, s^*) + \ln |A^T A|$ where $s^* = (A^T A)^{-1} A^T y$.

Synthetic displacements and tilts from a point dislocation source in a homogeneous half space are calculated following Okada[40]. Following Legrand

et al.[31], we set the P velocity of 1500 m/s, S-velocity of 800 m/s, and the density of 1700 kg/m³, respectively. To evaluate the significance of the single force component, we also calculate synthetic displacement and tilt from a point force source following equations 1–2 in Okada[40]. To consider the station elevation (i.e., typically 600 m above sea level), we set the station elevation as the free surface and the depth of the source is adjusted accordingly in the calculation of synthetics. Since the borehole tiltmeter is typically about 100 m below the surface, we calculate synthetic tilts at the borehole depth.

We first use differential evolution method[75] to find the global minimum of $-\ln p(\boldsymbol{m}, \boldsymbol{s}, \boldsymbol{\beta}|\boldsymbol{d})$. This step is repeated for hundreds of times with the randomly assigned initial values. A consistently converged minimum is regarded as the global minimum. Subsequently, we use an ensemble Monte Carlo Markov Chain method[76] to sample the posterior probability density function. The starting values of ensemble Markov chains of $k^{th}$ parameter are randomly assigned according to the global minimum by a normal distribution, $N(x_k, |x_k|)$, where $x_k \in \boldsymbol{m}, \boldsymbol{s}, \boldsymbol{\beta}$. This step is also repeated for multiple times to evaluate the convergence of independent Markov chains. Finally, the posterior probability density functions of the parameters are combined from multiple independent chains (Supplementary Fig. 12). The moment-tensor representation of the source can be reconstructed from the moments of strike-slip, dip-slip, tensile crack, and explosion[40], and described in terms of the fundamental lune[77].

**Estimating volume change**. As the strike-slip and dip-slip components do not experience any volume changes, the volume change associated with the reference event is estimated by only considering the tensile crack and explosion components. Following Okada[40], the eigenvalue of tensile-crack moment tensor is $\Lambda_{tc} = m_{tc}[1, 1, 1 + 2\mu/\lambda]^T$ and the eigenvalue of explosion moment tensor is $\Lambda_{ex} = m_{ex}[1, 1, 1]^T$. Following equation 16 in Kawakastu and Yamamoto[17], the volume change associated with the tensile crack component is $\Delta V_m^{tc} = m_{tc}/\lambda$ where $\lambda$ is Lame constant. Following equations 10–11 in Kawakastu and Yamamoto[17], the volume change associated with the explosion component in the presence of the confining pressure of the surrounding elastic medium is $\Delta V_m^{ex} = m_{ex}/(\lambda + 2\mu)$ where $\mu$ is the shear modulus. Hence, the 'Mogi volume' change associated with the reference event is $\Delta V_m = \Delta V_m^{ex} + \Delta V_m^{tc}$. Following Legrand et al.[31] and assuming $\lambda = 1.649\, GPa$ and $\mu = 1.088\, GPa$, $\Delta V_m = \sim 25840 \pm 6070\, m^3$ (Supplementary Table 2, a 4th-order polynomial function).

Following the note in section 5.1, the Mogi volume change corresponding to a 1-nrad tilt at N.ASHV.LE is ~440 m³. However, unlike the reference event, the static tilt offset in the monthly stacks can be difficult to measure directly in the time domain. As the duration of the tilt offset is typically ~100 s or less, the amplitude of the tilt-rate spectra plateaus in the near-field at the very long-period (i.e., >1000 s), corresponding to the amplitude of the tilt (Supplementary Fig. 13). We zero-pad the stacked waveform 10,000 s before the start of the tilt-rate time series and compute the tilt-rate amplitude spectra. The amplitude at the period of 10,000 s is used as a proxy for the static offset, which is scaled against the tilt of the reference event to obtain the volume change per event associated with each monthly stack. The monthly volume change can be obtained by multiplying the event number against the volume change per event (Supplementary Fig. 14).

**The effect of magma compressibility on the estimate of volume change**. Depending on the magma compressibility and chamber wall compliance[78], it is conceivable that we may underestimate the volume change by a factor of $R_\nu$, where $R_\nu = 1 + \frac{4}{3}\mu\chi$. Here $\mu$ is the wall rock rigidity and $\chi$ is the magma compressibility. Following Henry's law and the ideal gas law[78], the compressibility of fluid-gas magma is related to its pressure-dependent density $\rho_m$ by $\chi = \frac{1}{\rho_m} \times \frac{d\rho_m}{dP}$ and

$$\rho_m(P) = \left[ \frac{(\phi_0 - sP^n)RT}{PM_m} + \frac{1 - \phi_0 + sP^n}{\rho_0(1 + P\chi_l)} \right]^{-1},$$ where $P$ is the pressure; $T$ is the temperature; $\phi_0$ is the weight fraction of gas phase at atmospheric pressure; $s$ is the solubility of gas phase; $n$ is Henry's exponent of gas phase; $M_m$ is the molar mass of gas phase; $R$ is the ideal gas constant; $\rho_0$ is the density of liquid phase at atmospheric pressure; $\chi_l$ is the compressibility of the liquid phase. Assuming a gassed basaltic magma with $s = 5.9 \times 10^{-12}$ Pa $^{-1}$, $n=1$, $\rho_0 = 2650$ kg/m³, $\chi_l = 10\text{–}10$ Pa$^{-1}$, $T = 1000$ K, the chamber wall rock rigidity $\mu = 1\text{–}10$ GPa, and the gas weight percentage $\phi_0$ of 0.07% (at $P = 125$ MPa, see A1 magma in Saito et al.[21]), the estimated $R_\nu$ is 1.1–2.3.

**Estimate magma ascent velocity and decompression rate**. Following the 1-D conduit flow model by Jaupart & Tait[60], the conduit radius $R$ is balanced by the dynamic pressure of the ascended magma and lithospheric gradient $R^4 = G\frac{8\eta}{\pi\rho(\rho_r - \rho)g}$, where $G$ is the mass flow rate, $\eta$ is the magma viscosity, $\rho_r$ is the density of wall rock, $\rho$ is the density of magma, and $g$ is the gravitational acceleration. To estimate the conduit radius, the mass flow rate is estimated by the volume change rate of the inflation/deflation event and magma density $\rho$. The magma ascent velocity $V_a$ is estimated as $V_a = \frac{G}{\pi\rho R^2}$. Assuming a lithospheric gradient, the decompression rate $dP/dt$ is estimated as $dP/dt = \rho_r g V_a$.

## Data availabilty

The broadband and tilt waveform data can be accessed through https://www.hinet.bosai.go.jp. SO₂ emission is available from https://www.data.jma.go.jp/svd/vois/data/fukuoka/rovdm/Asosan_rovdm/gas/gas.html. Temperature data is available from http://www.data.jma.go.jp/svd/vois/data/tokyo/STOCK/monthly_v-act_doc/fukuoka/2016y/503_16y.pdf. As the VLP catalog remains the subject of several companion papers in preparation, it is available upon request.

## Code availabilty

The code computing synthetic deformation is available from https://www.bosai.go.jp/e/dc3d.html. The code estimating source parameters is available from https://doi.org/10.5281/zenodo.5082386. The code calculating mass flow rate and decompression rate is available from https://doi.org/10.5281/zenodo.5082382.

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

## Acknowledgements

We gratefully thank NIED and JMA for providing us the high-quality waveform data of V-net. We are also grateful to Prof. Mare Yamamoto and Prof. Takuhiro Ohkura for hosting J. Niu in Aso Volcanic Observatory and Tohoku University during the early stage of this work, which helps improve our understanding of the VLP and Aso volcanic system. The uses of the UCL Legion High-Performance Computing Facility (Legion@UCL), the UCL Grace High-Performance Computing Facility (Grace@UCL), the UCL Emerald High-Performance Computing Facility (Emerald@UCL), and associated support

services are acknowledged. J. Niu and T.-R. A. Song are supported by the Natural Environment Research Council, UK (NE/ P001378/1 & NE/T001372/1).

## Author contributions

J.N.: Conceptualization, Methodology, Software, Validation, Formal analysis, Investigation, Resources, Data curation, Writing—original draft, Visualization. T-R.A.S.: Conceptualization, Methodology, Validation, Formal analysis, Writing—review & editing, Supervision, Project administration, Funding acquisition, Conceptualization.

## Competing interests

The authors declare no competing interests.
