## [Peer Review File · Nature Communications]

REVIEWER COMMENTS

Reviewer #1 (Remarks to the Author):

Overall impression

The article written by Niu and Song shows an exciting interpretation of tons of tilt and seismic signals that repeatedly occurred at Aso volcano, Japan, with precise analysis procedures and its results. It sheds light on the new concept of the magma plumbing system at the volcano, adding a magma storage zone located between the magma chamber and the conduit. It identifies that this zone's inflation and deflation are associated with episodic magma discharge from below and are indeed relating to the surficial phenomena. After calculating the 1-D conduit flow, it improves the public image of magma ascent dynamics from the magma chamber to the surface. It also provides a future direction of geophysical research to real-time evaluate upcoming eruptions. This article seems to be well-organized, and the methods and results are detailed and very interesting for the scientific community widely. I would recommend their article for publishing in the journal after minor revisions of some comments below I pointed out.

Specific comments

P3 L13 - The authors use the term "long-period tremor (LPT)" for the objected seismic signal in the whole in this article because they respect the nomenclature that had been established in previous papers. However, this term seems to be a local vernacular or jargon only at Aso volcano. I am also suspect that that signal is not a "tremor?" I suggest using the more widely used term VLP or ultra-long-period (ULP) signals in the volcanological field instead of LPT for readers of this journal from more wide backgrounds.

P4 L4 - Were the VLP signals the authors checked from the catalog associated individually with any surficial eruption activities at the crater during the erupted periods? A paper reported that the occurrence of the VLP seismic signal preceded by few seconds from the onset of Strombolian eruption (Ishii et al., 2019, EPS). What the meaning of a difference of such elapsed time (2 min for Oct. 2016 phreatomagmatic eruption and 2-3 s for Strombolian explosions), as well as the meaning of a difference with or without eruptions? I think such a time difference may not relate to the patterns of the tilt source. Are there several types of VLPs with similar waveforms, but specific properties are different? Can the author comment on this topic at any part of this article?

P4 L20 - Can the authors show a result of the same method using another template (Event 2) in the supplementary? I could not find validity to use Event 1 (not 2) as a reference signal in this article. I am convincing that the conclusion will be the same as this article if Event 2 is used. However, curious about the time evolution of the VLPs for the case of 2.

P5 L15 - The authors should explain the excitation mechanism of the SPT concerning the occurrence of VLP as well as the series of phenomena starting from a deeper place (top of the magma chamber; the authors argue). I could not imagine why the SPT starts at the top portion of the conduit (VLP source) at first, and 10 s later, the VLP occurs (it is the same time as the SPT peak), whenever the events are inflation or deflation.

P6 L10 - The authors cited Hata et al. (2018, JGR) in this article; however, it seems inappropriate. At least another paper of Hata et al. (2018, JGR, 10.1029/2018JB015951), or much preferable Matsushima et al. (2020, EPS) showing the revised model of the Hata et al.'s result should be cited.

P6 L20 - Add any comment on how to make seismicity around the roof of the magma chamber if gas-dominant materials transport upward.

P7 L1 - Can the authors show other evidence of inflation and deflation during these periods? It would be enough to cite some papers of InSAR or GNSS that can strengthen their argument.

P7 L14 - I could not understand the concept relating to SO₂ gas emission. Could the author explain more carefully? The authors describe the prominence of inflation events equivalent to pressurization and lowering SO₂ emission. The relation between inflation and pressurization is readily accepted because both are relative

changes of the SMSZ condition. However, in my understanding, SO₂ emission does not seem to this relative change; but relates to the absolute volume of magma transported to a shallow depth. Is this wrong? I could not see a significant correlation between Fig 4b and 4c.

P9 L11 - Probably the wrong citation; did not Miyabuchi & Hara (2019, EPS) treat the 2016 phreatomagmatic explosion? I guess Ishii (2018, EPS), Ishii et al. (2018, EPS) and Sato et al. (2018, EPS) are more suitable for the mass of ejected materials.

P11 L 13 - Unclear to me.

References - It is not good to cite several papers written in Japanese, which most of the readers cannot probably reach and read. Other accessible papers should be referred to in this article as far as possible. The author also should discuss the proposed plumbing system with the result of Tsutsui & Sudo (2004, JVGR).

Fig. 2 - I guess the exact time of the onset in longer signals is quite tricky. How about the reading error?

P37 L 14 - need a reference for assumption (or evaluation)

Reviewer #2 (Remarks to the Author):

This paper provides interesting new observations of coupled seismic and ground deformation of repeated magma transport events observed at Aso volcano, Japan that are at least sometimes associated with eruptions. These types of high spatial and temporal resolution observations combining these datasets are rare (to my knowledge) and provide a unique perspective on the timing, location, and volume change of magma movements within a volcano. It would be good to get the perspective of an expert on the Aso system, but as far as I can tell, this paper provides new insight into the plumbing system at this volcano and is possibly applicable (in terms of the conceptual model and technique) to other volcanic systems.

I recommend publication after moderate revision. There are several steps in the analysis that are not well documented (see detailed comments below). Further, the paper is unclear in some locations or the discussion is incomplete (again documented below).

Page 1:

Line 10 and Page 1 lines 6-9: How do we know that Aso has a crystal rich mush and that it is relevant for this study? Maybe the magma batches are coming from a crystal poor reservoir? Maybe the crystal rich mush is deeper?

Line 14: "individual" should be "an individual"

Lines 16-24: I found the following sentences unclear and confusing -- can the authors be more specific? What do you mean by "composition dependent"? Is the composition of each magma batch different? Does the last sentence mean that you can forecast the eruption style, plume height, etc. based on the tilt/seismic data described above? If so provide some more details as to how.

"whereas their recurrences, potentially composition dependent, are regulated by the brittle-to-ductile transition rheology under low differential stress and high strain rate due to the surge of magma from below, regulating long-term volcanic output rate. The magma ascent velocity, decompression rates, and cumulative magma output deduced from the episodic deformation events before recent eruptions in Aso volcano are compatible with retrospective observations of the eruption style, tephra fallouts, and plume heights, promising real-time evaluation of upcoming eruptions."

Further, the results shown in Figures 4-6 aren't really described in the abstract.

Page 2:

Line 5: the ambient stress state also matters

Line 16: Also the ambient stress state matters -- seismicity will only occur where the rocks are near to failure. Magma can move aseismically if the stress state is not close to failure.

Page 3:

Line 8: "signal" should be "signals"

Line 10: How do we know this is a "shallow hydrothermal reservoir"?

Line 11: use "on" instead of "against"

Line 12 (and Page 1 line 12): Mentioning the source is near sea level is confusing, how far is this below the surface? It would be better to tell us the depth of the source beneath the surface (or at least tell us both pieces of information).

Line 16: Need some introduction to the eruptive cycle -- why was the time period 2011-2016 chosen? Why not a longer time period? Is the LPT only seen in this time period? Is this the only time period when the patterns described below occur? Or some other reason?

Line 17: "waveform" should be "waveforms"

Line 21: How do you know these LPT events are "anomalous"? Where do you define normal or background LPT activity?

Page 4:

Line 4: where do the displacement waveforms come from? Integration of the seismograms? If so, how?

Line 5: are these events associated with eruptions?

line 6: What does "east-down" mean? Does that mean tilt toward the east?

Line 8-9: This phrase could be more precise: "between the signal of LPT and the tilt offset". Perhaps: "between the LPT signal and the tilt offset at different stations and in the different components at the same station."

Line 17: Are all the LPT events associated with eruptions? Are there LPT events that aren't found by the matched filter? If so, what type of events do they represent?

The sections entitled "LPT and synchronous tilt/displacement offset" and "Discovery of the inflation/deflation event beneath Aso volcano" could be better organized.

It seems to be organized in a chronological manner instead of a logical description of what was discovered -- the first paragraph talks about the 2016 eruption, the next paragraph is about a manual search and the next paragraph is about a matched filter. Instead, why not just discuss the procedure (pointing to the Materials and Methods as needed) and then describe what you found? Maybe organize: this is what we analyzed, describe the 2016 events (including the variation in signals between stations) and then the global stack.

Page 5,

line 4: How many events are in the stack? Do the number of events vary in time in a systematic manner? What do the unstacked events look like?

Line 5: what does volcanic unrest mean here? Are there eruptions in 2011-2014 or are these events occurring without eruption? If so, that is strange -- why do these similar events occur sometimes with eruption and sometimes without?

Page 6:

Line 4: Make clear from the source what your forward model is in terms of source characteristics and elastic structure.

Line 15: What is the composition of the magma batch based on the eruption? One hypothesis is that this portion of magma is ascending because it has accumulated enough gas to become buoyant, so what is known about the gas compositions in the eruptions?

Line 16: How do we know that this is a crystal rich magma? Is this just a guess or is there some evidence from petrology about the source region?

Line 20: An alternative is that the magma is that there is no new injection of magma and it is cooling/crystallizing, accumulating gas at the top of the reservoir and then episodically having sufficient buoyancy to cause brittle failure. (This is a top-down instead of a bottom-up trigger for eruptions, see for example Girona, T., Costa, F., & Schubert, G. (2015). Degassing during quiescence as a trigger of magma ascent and volcanic eruptions. *Scientific reports*, 5(1), 1-7.

Page 7:

Line 18: This conceptual model is reasonable, but what is the evidence that there is a crystal-rich or crystal-poor mush? What is the petrological evidence for the percentage of crystals? Is it really > 50% in one reservoir and < 50% in the other?

Page 8:

Line 2: You should be clear that you have demonstrated this for a particular volcano during a particular time period and not imply this is a universal process: "the upward transport of magma/gas from the magma chamber toward the surface is a stepwise process in an episodic fashion"

Page 9: I'm glad to see the discussion of gas (finally), but what are the observations on degassing rate measured on the ground or by satellite? Is it really likely that some events have a high gas proportion and over events a few months later have a low gas proportion? Further, the volume discrepancy might not have anything to do with gas, but could be due to additional reservoirs being tapped that were filled long before the current eruption (that may not have a tilt/tremor signature). A volume difference of 6-8 is at the high range considered by Rivalta and Segall (2008) but maybe appropriate for this arc volcano? What do the authors think this ratio implies? The question of gas in the magma could be uniquely addressed with this dataset.

Page 11:

Line4: This sentence is filled with either assumptions or claims that aren't yet supported in the manuscript such as the existence of a "crystal-rich mush and crystal-poor pool"

Line 6: Is this claim discussed further somewhere? If so, I missed it: "The duration of each deformation event (~ 50 s) is much longer than what is expected for crustal earthquakes of similar size and such a slow deformation"

The perspective could be improved with adding a paragraph or two about how the lessons here could be applied to specific other volcanoes (if possible). Applying the techniques to other volcanoes is mentioned, but what other volcanoes have a similar eruptive style and might be the best targets to investigate? Also, there are many types of eruptions or plumbing systems for which these types of analysis would not work and should be mentioned as well. If additional space is needed, I suggest dropping Figs. 5 and 6 below.

The data availability statement is not clear: Are the tilt and seismic waveforms available from the link provided? Also the statement that data products are available by request is no longer considered a best practice (for example, it is not allowed by AGU). These data products are not required to be made available in a public repository, but it would add great value if they were. Does UCL have such a repository?

Fig. 1: What are BCU and BYA chambers? They are not mentioned in the caption. Also, what is the depth of the low velocity zone, and other features listed in the legend (maybe refer to Fig. 3)?

Fig. 2: Some more details are needed in the caption. How many events are stacked together here? How were events horizontally aligned?

Fig. 3: Where do the horizontal and vertical displacements in a and b come from? GNSS? Or integrated from seismometer? I could be helpful in d to show the depths of the features shown in a, b, and c: where are the low velocity zone, BYA and BCA chambers, inverted source location (Red Cross) and new Mogi (black circle)? Why is there an aquifer labeled in d? I don't think the aquifer is mentioned in the caption or the text. I'm also confused about what is happening in e, f, and g. What are the red and blue dotted lines at the line labeled LPT? What are the arrow at the SPT line? What are the arrows in-between the LPT and SPT lines? What physical processes do these arrows/features represent?

Fig. 4a: where does the accumulative net volume change come from? The data used to calculate this should be mentioned in the caption. I do not understand the labels that say, for example, $\Delta V_{\text{magma}} \ll \Delta V_{\text{gas}}$ during time period 1. It looks like the volume change is basically flat during this time period, so shouldn't these two volume changes be in approximate balance instead of orders of magnitude different (as implied by using \ll)?

Fig. 4b: What is "outgassing potential"? I haven't heard this term before and the phrase used in the caption is still confusing: "the moment ratio between pressurization and 10 depressurization LPTs"

Fig. 4d: Where is the volume change rate being measured? SMSZ? The caption says the black crosses are the "inflation event" but I think this should be "inflation events." What is the geodetic data used to estimate the green cross? Is this the same tilt data described in this paper or something else, like GNSS? What is the time period of the geodetic data? In general, it should be noted that these volume change rates are being measured over vastly different time periods and the time periods should be mentioned in the caption.

Fig. 4e: Considering there are only 2 data points, it does not seem wise to draw a line between them.

Figures 5 and 6 have minimal discussion in the text and do not seem to factor into the key conclusions mentioned in the abstract -- could they be removed? They probably deserve further discussion in a separate paper. Fig. 5 has a huge amount of information that isn't discussed in the main text. In particular, I think Fig. 6 is confusing and possibly misleading. It seems to take a single volcanic system and wildly extrapolate it to all systems worldwide. This figure seems to imply that the magma plumbing system of Aso is relevant to all types of eruptions from Rhyolites to flood basalts which is clearly not true -- we have enough information to know that the plumbing of Aso is not widely applicable to all volcanoes. The authors should consider what is the point this figure is trying to make and if it is already made successfully in the text. If the point is that "Composition, viscosity, rheology and tectonic settings govern the recurrences of episodic

deformation" then this point can be made adequately in the text without confusing the reader.

Fig. 6: "providing the glue" is a confusing phrase here

Reviewer #3 (Remarks to the Author):

Review comments on Episodic transport of discrete magma batches beneath Aso volcano
By Jieming Niu, Teh-Ru Alex Song

This manuscript is quite interesting and is sufficiently valuable to publish on Nature Communications.

Major comments

- 1) Source of tilt offset is discussed in the relation of crystal rich and crystal poor zones. I cannot well understand how to relate the source of tilt off set with the rate of crystal. The reference 48 investigate volatile from the viewpoint of petrology. If source of tilt offset is related with volatile-rich and volatile-poor zones, this might be better understood.
- 2) Source of tilt offset is also discussed on brittle-ductile transition zone. However, the authors assumes that tilt offset is induced by volume change of a combination model of tensile crack and explosive source in elastic medium and this model does not include fracture. Brittle and ductile are manners of fracture. Regarding to the model of elastic deformation, source of tilt offset should be discussed on difference in elastic constants along the magma plumbing system.
- 3) The significance of comparison of 2011-2016 eruptivity of Aso with basaltic eruptivity is not well understood. As mentioned in the text, long-term eruption rate of andesitic volcanoes is lower than basaltic volcanoes. I cannot find a significance of comparison of eruption rate between andesitic and basaltic volcanoes in this manuscript. The eruption 2011-2016 is an eruptive activity of Aso, however it does not cover all the eruptivity of Aso. If compared, the eruption 2011-2016 should be compared with past eruptivity of Aso or long-term eruptivity of the volcano.

Minor comments

P3L20

"natural period" -> band width

P4L12

LP signal east-west is much weaker, but north-south is stronger than N.ASIV.

P4L13 "These observations strongly indicate that the source of the tilt offset is spatially separated from the LPT source"

It is possible, but is it necessary to examine source difference between tilt offset and LPT?

P4L14 "which is near the active Naka-dake first crater"

Show references or see Method.

P5L3 global waveform stacks

What do you mean "global"? How many LPTs were stacked?

P5L9 relatively steady

Almost the same?

P5L21 Fig. S4

Fig. S4 is "Synthetic amplitude-distance decay against static and filtered waveforms". Inserting Fig. S4 explains why you choose 100-200s ULP band?

P7L2 "SMSZ"

"SMSZ" firstly appeared here. This should be explained.

P7L5 intense

Is this word necessary? What do you mean "intense unrest"?

P7L18 "a deeper reservoir (i.e., a crystal-rich mush) and temporarily stalled in the SMSZ (i.e., a crystal-poor pool)"

The reference 48 investigate volatile. How does it relate to crystal? Are there any evidence for crystal-poor in the SMSZ?

P8L1-L7

For estimation of volume increase, Method should be referred.

P8L13-L22

This hypothesis is true. As the authors show clearly, LPT is a short-lived phenomenon. What does the author mention, comparing short-lived volume change with long-term eruptivity?

P10L11-L13 "While the estimated mass flow rate in Aso is lower than those estimated in basaltic eruptions by an order of magnitude (Fig. 5), such a difference is consistent with the disparity in the average volcanic output rate between basaltic eruptions and andesitic eruptions"

In this case, Aso means activity 2011-2016? Which eruption does "basaltic eruption" indicate?

The first half is comparison of short term activity. Second half is long-term comparison. Long-term comparison may be true. But short-term comparison is case-by-case. It is not necessary to be consistent.

P22L11

Takuhiro ->Takahiro

Figures

P24L9 October 8, 2016

Need time of onset of the eruption.

(b) N.ASHV.LE is tilt of east-side down at station N.ASHV?

Is Figure 6 needed?

In the following response, the comments by the reviewers are shown in *italic*, our responses are shown in **bold**. The page and line numbers in the original manuscript are noted as **P?L??**. The page and line numbers in the revised manuscript are noted as **nP?L??**. The responses discussed in the rebuttal letter are also highlighted in yellow in the revised manuscript.

Reviewer #1 (Remarks to the Author):

Overall impression

The article written by Niu and Song shows an exciting interpretation of tons of tilt and seismic signals that repeatedly occurred at Aso volcano, Japan, with precise analysis procedures and its results. It sheds light on the new concept of the magma plumbing system at the volcano, adding a magma storage zone located between the magma chamber and the conduit. It identifies that this zone's inflation and deflation are associated with episodic magma discharge from below and are indeed relating to the surficial phenomena. After calculating the 1-D conduit flow, it improves the public image of magma ascent dynamics from the magma chamber to the surface. It also provides a future direction of geophysical research to real-time evaluate upcoming eruptions. This article seems to be well-organized, and the methods and results are detailed and very interesting for the scientific community widely. I would recommend their article for publishing in the journal after minor revisions of some comments below I pointed out.

We thank the reviewer for the positive comments and support for the publication.

Specific comments

P3 L13 - The authors use the term “long-period tremor (LPT)” for the objected seismic signal in the whole in this article because they respect the nomenclature that had been established in previous papers. However, this term seems to be a local vernacular or jargon only at Aso volcano. I am also suspect that that signal is not a “tremor?” I suggest using the more widely used term VLP or ultra-long-period (ULP) signals in the volcanological field instead of LPT for readers of this journal from more wide backgrounds.

We thank the reviewer for the suggestion.

Indeed, the LPT is associated with event-like signal, and it does not resemble typical tremors. We have replaced LPT with the more widely recognized term VLP in the revised manuscript. We also replaced the term SPT with the more widely recognized term LP to represent the long-period signal above the crack-like conduit in Aso volcano.

P4 L4 - Were the VLP signals the authors checked from the catalog associated individually with any surficial eruption activities at the crater during the erupted periods?

We thank the reviewer for the comment.

Some VLPs are indeed associated with individual surficial eruption during the erupted periods. This corroborates the observation by Ishii et al. (2019, EPS). We briefly noted this in the revised manuscript (nP11L6).

A paper reported that the occurrence of the VLP seismic signal preceded by few seconds from the onset of Strombolian eruption (Ishii et al., 2019, EPS). What the meaning of a difference of such elapsed time (2 min for Oct. 2016 phreatomagmatic eruption and 2-3 s for Strombolian explosions), as well as the meaning of a difference with or without eruptions? I think such a time difference may not relate to the patterns of the tilt source.

We thank the reviewer's comment.

Motivated by the discussions by Ishii et al. (2019), we have added two paragraphs and highlighted the implications of these differences against the inferred magma ascent velocity

in the SMSZ (nP11L3-L13). We conceive that the time delay between VLP and the onset of Strombolian eruption can be reasonably attributed to gas ascent in the shallow conduit under slow magma ascent (e.g., 0.01-0.1 m/s). On the other hand, the 120 s time delay between Event 2 and the onset of 2016 phreatomagmatic eruption can be reconciled with the relatively magma ascent in the SMSZ (e.g., ~ 7 m/s).

On the other hand, VLP and synchronous inflation/deflation event does occur with or without eruption. It implies that the occurrence of VLP is likely a necessary condition for surface eruption. As discussed by Niu & Song (2020), conduit/plug permeability and overpressure also play a key role in facilitating surface eruptive activities. We have added a short note in the revised manuscript (nP8L20-L24).

Are there several types of VLPs with similar waveforms, but specific properties are different? Can the author comment on this topic at any part of this article?

We thank the reviewer's comment.

We have commented their differences (e.g., resonance period, initial polarity) in the revised manuscript (nP3L10-L14).

P4 L20 - Can the authors show a result of the same method using another template (Event 2) in the supplementary? I could not find validity to use Event 1 (not 2) as a reference signal in this article. I am convincing that the conclusion will be the same as this article if Event 2 is used.

We thank the reviewer for the comments.

Choosing Event 1 as the template has one advantage in that it potentially minimises the interference from the eruption signal at long period (i.e., 50-250 sec). However, the waveforms from Event 1 and Event 2 are highly correlated, using Event 2 as the reference event does not result in noticeable difference in the detection of inflation or deflation events. The inverted source location and mechanism from Event 2 are practically the same as those inverted from Event 1 (see figure below). We do not include the inversion result from Event 2 in this paper to avoid redundancy. We emphasized the high correlation between Event 1 and Event 2 in the revised manuscript (nP3L24-P4L1).

Rebuttal Figure 1 Posterior probability functions of the inverted source parameters from Event 2. Notations are the same as Supplementary Figure 12 in the manuscript.

However, curious about the time evolution of the VLPs for the case of 2.

We thank the reviewer's comment.

Perhaps there is a misunderstanding here. We emphasize that waveforms from Event 1 and Event 2 are practical identical in the period of our interest (i.e., 50-250 sec). Therefore, VLP signal of 10 sec period is not the focus of our detection. Rather, the timing of VLP was used to detect inflation or deflation events.

In a separate note, the evolution of VLPs has been thoroughly documented by Niu & Song (2020, JVGR) and we refer the reader to this article for details.

P5 L15 - The authors should explain the excitation mechanism of the SPT concerning the occurrence of VLP as well as the series of phenomena starting from a deeper place (top of the magma chamber; the authors argue). I could not imagine why the SPT starts at the top portion of the conduit (VLP source) at first, and 10 s later, the VLP occurs (it is the same time as the SPT peak), whenever the events are inflation or deflation.

We thank the reviewer for this comment.

Following the discussions by Kaneshima et al. (1996) and Kawakatsu et al. (2000), we have added a paragraph to note the possible processes associated with VLP and LP. We emphasize the stress perturbation induced during the initial stage of the inflation/deflation event results in fractures in the conduit plug above the shallow crack-like conduit, promoting LP and VLP (nP6L7-L12).

We have appended additional paragraphs in the revised manuscript and elaborated the series of phenomena starting from the deep source near the chamber roof (nP7L3-L28). We caution that this is a tentative interpretation at this point and a more elaborated discussion is beyond the main scope of this manuscript.

P6 L10 - The authors cited Hata et al. (2018, JGR) in this article; however, it seems inappropriate. At least another paper of Hata et al. (2018, JGR, 10.1029/2018JB015951), or much preferable Matsushima et al. (2020, EPS) showing the revised model of the Hata et al.'s result should be cited.

We thank the reviewer for pointing out this issue.

We have cited Hata et al. (2018, JGR, 10.1029/2018JB015951) and Matsushima et al. (2020, EPS) in the revised manuscript (nP6L23).

P6 L20 - Add any comment on how to make seismicity around the roof of the magma chamber if gas-dominant materials transport upward.

We thank the reviewer for this comment.

As also included in the reply earlier, we noted the role of gassed magma in the inflation/deflation events in the revised manuscript (nP7L7-L13). The discussions are in line with the suggestion by the reviewer3 that gas near at the top of the magma chamber may facilitate episodic brittle failure due to increasing magma buoyancy.

P7 L1 - Can the authors show other evidence of inflation and deflation during these periods? It would be enough to cite some papers of InSAR or GNSS that can strengthen their argument.

We thank the reviewer for the comment.

As noted in the original manuscript, the signals associated with the inflation or deflation events are on the order of 1 μ m in displacement, which is much smaller than the detection threshold of InSAR and GNSS (i.e., mm to cm). We have emphasized the limitation in the revised manuscript (nP2L22-L28).

Furthermore, as readily discussed in the Methods sections, to identify the displacement or tilt offset, we remove the mean of the background trend before the LPT arrival. As suggested in the original manuscript, these detected events likely represent episodic transport of discrete magma from the roof of the magma chamber to the storage zone directly above, i.e., the source and sink are probably too close to be deciphered from GNSS or InSAR.

On the other hand, we note that GNSS displacement from JMA show a notable inflation of the magma chamber in mid-July 2014, May 2015 and July 2016, suggesting magma ascent from a deep reservoir ($> \sim 10$ km) toward the magma chamber. These episodes coincide with a substantial increase in the number of inflation events and magma transport toward the SMSZ. We have briefly discussed this in the revised manuscript (nP8L12-L14).

P7 L14 - I could not understand the concept relating to SO₂ gas emission. Could the author explain more carefully? The authors describe the prominence of inflation events equivalent to pressurization and lowering SO₂ emission. The relation between inflation and pressurization is readily accepted because both are relative changes of the SMSZ condition. However, in my understanding, SO₂ emission does not seem to this relative change; but relates to the absolute volume of magma transported to a shallow depth. Is this wrong? I could not see a significant correlation between Fig 4b and 4c.

We thank the reviewer for this comment.

In the original manuscript, we refer to the observation that the prominence of inflation (deflation) events coincides with a higher proportion of the pressurization (depressurization) VLP event in the crack like conduit and a lower (higher) SO₂ emission. We do not mean to suggest that there is causal relationship between SO₂ emission and the prominence of inflation events. As discussed in Niu & Song (2020), the outgassing potential shown in Fig. 4b in the original manuscript was meant to infer conduit/plug permeability. Therefore, we do not necessarily expect a high correlation between Fig. 4b and Fig. 4c in the original manuscript.

On the other hand, we do observe a clear correlation between rising SO₂ emission, the accumulative volume of magma transport in the SMSZ and crater bottom temperature before the 2014 Strombolian eruption. We have replaced Fig 4b with the crater bottom temperature (now Fig. 5b in the revised manuscript) highlight these observations in the revised manuscript (nP8L14-L18).

P9 L11 - Probably the wrong citation; did not Miyabuchi & Hara (2019, EPS) treat the 2016 phreatomagmatic explosion? I guess Ishii (2018, EPS), Ishii et al. (2018, EPS) and Sato et al. (2018, EPS) are more suitable for the mass of ejected materials.

We thank the reviewer's comment.

Miyabushi & Hara (2019, EPS) concerned about the eruptive mass of the 2014 strombolian eruption, which is not a suitable citation. On the other hand, the reference cited in the original manuscript was Miyabuchi et al. (2017, JpGU abstract), which reported the eruptive mass from the field survey. We note the result of field survey by Miyabuchi et al., (2017) has also been cited by Ishii (2018), Ishii et al. (2018) and Sato et al. (2018). To keep the reference list not too long, we have cited Ishii (2018, EPS) in the revised manuscript (nP9L27).

P11 L 13 - Unclear to me.

We thank the reviewer for the comment.

We have rephrased and clarified how one may use the amplitude scaling between VLP and the inflation (or deflation) event derived in recent eruptions to evaluate pre-eruptive volume change in the SMSZ that is otherwise inaccessible for historical eruptions without modern data (nP12L11-L16).

References - It is not good to cite several papers written in Japanese, which most of the readers cannot probably reach and read. Other accessible papers should be referred to in this article as far as possible.

We thank the reviewer's comment.

There are four cited papers written in Japanese. The reference (54,69) in the original manuscript have been removed. However, the results in references (40,49) are essential and there is no alternative literature published in English. Therefore, we retain the two references (40, 49) (reference 46, 45 in the revised manuscript).

40. Ohkura et al. (2009).

49. Sudo et al. (2006).

54. Yokoo & Miyabuchi (2015).

69. Sakaguchi et al. (2008).

The author also should discuss the proposed plumbing system with the result of Tsutsui & Sudo (2004, JVGR).

We thank the reviewer's comment.

We have cited Tsutsui & Sudo (2004) in the proposed plumbing system in the revised manuscript (nP6L20).

Fig. 2 - I guess the exact time of the onset in longer signals is quite tricky. How about the reading error?

We thank the reviewer for this comment.

The onset of the long-period signal is identified when the velocity exceeds the peak-to-peak amplitude of the background noise (Fig. 2). We estimate the picking error of ~ 5 sec or less.

P37 L 14 - need a reference for assumption (or evaluation)

We thank the reviewer for the suggestion.

There was a typo in the Lamé's constant in the original manuscript (P37L14). The correct Lamé's constant λ is 1.649 GPa. Following Legrand (2000), we assume $v_p = 1500$ m/s, $v_s = 800$ m/s and $\rho = 1700$ kg/m³. We have cited this reference in the supplementary. Since the period of interest is 50 sec and longer, as detailed in the reply to the reviewer 3, the absolute value of the elastic constant does not change the source characteristics or the estimated volume change. We noted this in the revised manuscript (nP5L26-L29).

Reviewer #2 (Remarks to the Author):

This paper provides interesting new observations of coupled seismic and ground deformation of repeated magma transport events observed at Aso volcano, Japan that are at least sometimes associated with eruptions. These types of high spatial and temporal resolution observations combining these datasets are rare (to my knowledge) and provide a unique perspective on the timing, location, and volume change of magma movements within a volcano. It would be good to get the perspective of an expert on the Aso system, but as far as I can tell, this paper provides new insight into the plumbing system at this volcano and is possibly applicable (in terms of the conceptual model and technique) to other volcanic systems.

We thank the reviewer for the positive comments and support for the publication.

I recommend publication after moderate revision. There are several steps in the analysis that are not well documented (see detailed comments below). Further, the paper is unclear in some locations or the discussion is incomplete (again documented below).

Page 1:

Line 10 and Page 1 lines 6-9: How do we know that Aso has a crystal rich mush and that it is relevant for this study? Maybe the magma batches are coming from a crystal poor reservoir? Maybe the crystal rich mush is deeper?

We thank the reviewer for the comment, which help improve the interpretation.

We largely agree with the reviewer's suggestion. In the revised manuscript, the abstract has been modified. We note that the crystal-rich mush domain likely remains at a deeper depth (i.e., >10 km) after the most recent Aso-4 caldera forming eruption (e.g., Ishibashi et al., 2018) (nP6L13-L18).

Line 14: "individual" should be "an individual"

We thank the reviewer for the comment.

It has been removed in the abstract in the revised manuscript.

Lines 16-24: I found the following sentences unclear and confusing -- can the authors be more specific? What do you mean by "composition dependent"? Is the composition of each magma batch different? Does the last sentence mean that you can forecast the eruption style, plume height, etc. based on the tilt/seismic data described above? If so provide some more details as to how.

"whereas their recurrences, potentially composition dependent, are regulated by the brittle-to-ductile transition rheology under low differential stress and high strain rate due to the surge of magma from below, regulating long-term volcanic output rate. The magma ascent velocity, decompression rates, and cumulative magma output deduced from the episodic deformation events before recent eruptions in Aso volcano are compatible with retrospective observations of the eruption style, tephra fallouts, and plume heights, promising real-time evaluation of upcoming eruptions."

We thank the reviewer's comment.

First, we clarify the misunderstanding that the composition of each magma batch is different. Rather, "composition-dependent" was referred to the hypothesis that the recurrence interval of magma transport in a more silicic magma may occur much less frequently than that in a basaltic or andesitic magma. We have removed this sentence in the revised manuscript to avoid confusion.

The last sentence in the abstract in the original manuscript does imply the possibility of assessing the eruption size and style through the estimated mass flow rate and magma ascent velocity before the upcoming eruption. While the abstract has been rewritten, we simply note the effect of magma composition on the recurrence interval of episodic deformation events (nP10L12-L14).

Further, the results shown in Figures 4-6 aren't really described in the abstract.

We thank the reviewer for the comment.

As suggested by the reviewers, we removed Figure 6 in the original manuscript. The results in Figs. 4-5 (Figs. 5-7 in the revised manuscript) are honoured in the abstract in the revised manuscript.

Page 2:

Line 5: the ambient stress state also matters

We thank the reviewer for this comment.

We have modified the sentence accordingly in the revised manuscript (nP2L4).

Line 16: Also the ambient stress state matters -- seismicity will only occur where the rocks are near to failure. Magma can move aseismically if the stress state is not close to failure.

We thank the reviewer for this comment.

We agree with the reviewer and have stressed this point in the revised manuscript (nP2L18-L19).

Page 3:

Line 8: "signal" should be "signals"

We thank the reviewer for this comment.

We have modified the phrase accordingly (nP3L4).

Line 10: How do we know this is a "shallow hydrothermal reservoir"?

We thank the reviewer for the comment.

The presence of a shallow hydrothermal reservoir was inferred from a high electrical conductivity channel in several literatures (Hase et al., 2005; Hata et al., 2016; 2018; Kanda et al., 2008, 2019). We have cited relevant references in the revised manuscript (nP3L7-L9).

Line 11: use "on" instead of "against"

We thank the reviewer for this comment.

This phrase has been modified accordingly (nP3L7).

Line 12 (and Page 1 line 12): Mentioning the source is near sea level is confusing, how far is this below the surface? It would be better to tell us the depth of the source beneath the surface (or at least tell us both pieces of information).

We thank the reviewer for the comment.

We have appended the source depth in the revised manuscript (nP3L9).

Line 16: Need some introduction to the eruptive cycle -- why was the time period 2011-2016 chosen? Why not a longer time period?

We thank the reviewer for this comment.

The typical eruption cycle in Aso volcano has been summarized previously (e.g., Sudo et al., 2006, Kawakatsu et al., 2000). We have added some details in the introduction (nP3L15-L20).

Analysis of a larger dataset including activities after 2016 is currently undertaking and will be reported in the future.

Is the LPT only seen in this time period?

We thank the reviewer's comment.

As stated in the original manuscript, LPT can be observed regardless of surface activity and it has also been observed in the past (e.g., Sassa, 1933; Kaneshima et al., 1996; Kawakastu et al., 2000; Niu & Song, 2020). As noted above, the analysis includes a relatively complete Aso eruption cycle in 2011-2016.

Is this the only time period when the patterns described below occur? Or some other reason?

As noted above, the analysis and detection of inflation/deflation events includes a relatively complete Aso eruption cycle in 2011-2016. Analysis of a larger dataset beyond 2016 is currently undertaking and will be reported in the future.

Line 17: "waveform" should be "waveforms"

We thank the reviewer for this comment.

We have modified accordingly in the revised manuscript (nP3L23).

Line 21: How do you know these LPT events are "anomalous"? Where do you define normal or background LPT activity?

We thank the reviewer for this comment.

The LPT catalog has been detailed in Niu & Song (2020, JVGR). The amplitude of the two LPT events prior to the 2016 eruption is at least 2 orders magnitude larger than any LPTs we identified in 2011-2016. We add a brief note in the revised manuscript (nP3L28-nP4L1).

Page 4:

Line 4: where do the displacement waveforms come from? Integration of the seismograms? If so, how?

We thank the reviewer for the comment.

The vertical and horizontal displacement are integrated from the seismograms. This signal processing has been described in Methods in the original manuscript and in the revised manuscript.

Line 5: are these events associated with eruptions?

We thank the reviewer for this comment.

As shown in Fig S1, these events are not necessarily associated with eruptions. We have noted this in the revised manuscript (nP4L16).

line 6: What does "east-down" mean? Doe that mean tilt toward the east?

We thank the reviewer for the comment.

Yes, "east-down" tilt means that tilt toward the east. We have rephrased the sentence in the revised manuscript (nP4L13-L14).

Line 8-9: This phrase could be more precise: "between the signal of LPT and the tilt offset". Perhaps: "between the LPT signal and the tilt offset at different stations and in the different components at the same station."

We thank the reviewer for the suggestion.

We have rephrased accordingly in the revised manuscript (nP4L17-L21).

Line 17: Are all the LPT events associated with eruptions?

We thank the reviewer for this comment.

As detailed in P3L16 in the original manuscript, LPT is repetitive regardless of surface activity. Hence, they are not necessarily associated with eruptions.

Are there LPT events that aren't found by the matched filter? If so, what type of events do they represent?

The detection capability of the matched filter has been detailed in Niu & Song (2020, JVGR) and they have shown a nominal missed pick rate ~0.1%. When the signal-to-noise ratio is low, the matched filter will not be able to detect or characterize very small LPT events.

The sections entitled "LPT and synchronous tilt/displacement offset" and "Discovery of the inflation/deflation event beneath Aso volcano" could be better organized.

It seems to be organized in a chronological manner instead of a logical description of what was discovered -- the first paragraph talks about the 2016 eruption, the next paragraph is about a manual search and the next paragraph is about a matched filter. Instead, why not just discuss the procedure (pointing to the Materials and Methods as needed) and then describe what you found? Maybe organize: this is what we analyzed, describe the 2016 events (including the variation in signals between stations) and then the global stack.

We thank the reviewer for this comment.

We have reorganized this section in the revised manuscript (nP3L22-P5L21). Fig. S4 in the original manuscript has now been moved to Fig. S1 in the revised manuscript.

Page 5,

line 4: How many events are in the stack?

We thank the reviewer for this missing information.

There are 671 and 967 events for the global inflation and deflation stacks, respectively. This is appended in the caption of the revised manuscript.

Do the number of events vary in time in a systematic manner?

The number of events is relatively low in 2011-2013. The activity increases notably in early 2014 and substantially since July 2014. The monthly event numbers are included in the Fig. S5-10.

What do the unstacked events look like?

Waveforms of the unstacked events have been shown in Fig. S1 in the original manuscript and in the revised manuscript.

Line 5: what does volcanic unrest mean here?

We thank the reviewer for this comment.

The volcanic unrest includes surface volcanic activities such as the dried-up of the crater-late, minor phreatic and ash eruptions and incandescent phenomena. We have appended a sentence in the revised manuscript (nP4L27-L28).

Are there eruptions in 2011-2014 or are these events occurring without eruption? If so, that is strange -- why do these similar events occur sometimes with eruption and sometimes without?

As noted in Fig. 4, there are isolated minor phreatic eruption and ash eruptions in 2011-2014, but the detected events are generally not associated with eruptions. As noted by Niu & Song (2020), surface eruptions are not only dictated by the overpressure (i.e., magma

supply), but also the strength/permeability of the conduit plug. We have appended a brief note in the revised manuscript (nP8L20-L24).

Page 6:

Line 4: Make clear from the source what your forward model is in terms of source characteristics and elastic structure.

We thank the reviewer for this comment.

While the details of the forward model are described in Method of the initial submission, we add a sentence in the revised manuscript to clarify the source characteristics and elastic structure (nP5L26-L27).

Line 15: What is the composition of the magma batch based on the eruption? One hypothesis is that this portion of magma is ascending because it has accumulated enough gas to become buoyant, so what is known about the gas compositions in the eruptions?

We thank the reviewer for the comment.

The composition of the magma batch is basalt-andesitic, and the detailed analysis has been performed by Saito et al. (2018). They estimated that the buoyancy associated with the gassed magma is up to ~150-350 g/cm³, facilitating magma ascent. We have noted the role of magma buoyancy in the revised manuscript (nP7L7-L12).

Line 16: How do we know that this is a crystal rich magma? Is this just a guess or is there some evidence from petrology about the source region?

We thank the reviewer for this comment.

We refer to earlier response in the rebuttal letter. We also briefly note the crystal-rich mush region in the revised manuscript (nP6L13-L18).

Line 20: An alternative is that the magma is that there is no new injection of magma and it is cooling/crystallizing, accumulating gas at the top of the reservoir and then episodically having sufficient buoyancy to cause brittle failure. (This is a top-down instead of a bottom-up trigger for eruptions, see for example Girona, T., Costa, F., & Schubert, G. (2015). Degassing during quiescence as a trigger of magma ascent and volcanic eruptions. Scientific reports, 5(1), 1-7.

We thank the reviewer for the comment.

We agree that accumulating gas at the top of the reservoir may facilitate episodic brittle failure and we have included this mechanism in the revised manuscript (nP7L7-L19).

As noted in the rebuttal letter earlier, GNSS data reported by JMA indicate a slowdown of deflation in the magma chamber in July 2014, May 2015 and July 2016, suggesting a notable injection of magma from a deeper reservoir below toward the bottom of the magma chamber. These episodes correspond to increasing activities of inflation events and rising crater bottom temperature and SO₂ emission (Fig. 5). The discussions are added in the revised manuscript (nP8L12-L18).

Page 7:

Line 18: This conceptual model is reasonable, but what is the evidence that there is a crystal-rich or crystal-poor mush? What is the petrological evidence for the percentage of crystals? Is it really > 50% in one reservoir and < 50% in the other?

We thank the reviewer for this comment.

We note that the crystal-rich mush domain likely remains at a deeper depth (i.e., >10 km) after the most recent Aso-4 caldera forming eruption (Ishibashi et al., 2018). On the other hand, we have revisited the magma plumbing system recently summarised by Kawaguchi et al. (2021) and modified the interpretation in the revised manuscript. Specifically, we noted

the process of magma mixing for post-caldera volcanisms beneath Aso volcano (Miyoshi et al. 2011; Miyoshi et al., 2012; Kawaguchi et al., 2021) and inferred discrete magma transport between a chamber with volatile-poor silicic magma and a storage zone of mixed-magma (i.e., SMSZ) (nP6L13-L19).

Page 8:

Line 2: You should be clear that you have demonstrated this for a particular volcano during a particular time period and not imply this is a universal process: "the upward transport of magma/gas from the magma chamber toward the surface is a stepwise process in an episodic fashion"

We thank the reviewer's comment. We have rephrased the sentence in the revised manuscript (nP8L20-L22).

Page 9: I'm glad to see the discussion of gas (finally), but what are the observations on degassing rate measured on the ground or by satellite?

We thank the reviewer's comment.

The observations of SO₂ emission were from the campaign ground-based sensor and it is noted in the Fig. 5 caption in the revised manuscript.

Is it really likely that some events have a high gas proportion and over events a few months later have a low gas proportion?

We thank the reviewer's comment.

As detailed later in the rebuttal letter, we stress that the tilt-offset in some of the deflation events during the eruption may decay beyond the time scale of our analysis (i.e., $> \sim 1$ hr) and the volume change of these events is likely associated with the transport of gas, rather than magma. We suspect that such deflation events are likely of a small magnitude and lower signal-to-noise ratio.

We have appended the net volume change estimated from events with a high signal-to-noise ratio (i.e., subset I+II) (Fig. 5a). While the estimated net volume change before the eruption remains the same, the net volume change in the SMSZ during the eruption is more compatible with the net volume change before the eruption. We have expanded and clarified the discussions in the revised manuscript (nP9L11-L24).

Further, the volume discrepancy might not have anything to do with gas, but could be due to additional reservoirs being tapped that were filled long before the current eruption (that may not have a tilt/tremor signature). A volume difference of 6-8 is at the high range considered by Rivalta and Segall (2008) but maybe appropriate for this arc volcano?

What do the authors think this ratio implies? The question of gas in the magma could be uniquely addressed with this dataset.

We thank the reviewer's comment.

First, we followed Rivalta & Segall (2008) and included the effect of magma compressibility, on the estimate of volume change in SMSZ (Fig. 6a in the revised manuscript). The details are included in the Methods section. In short, we obtain $R_v=1.1-2.3$, which means that the eruption output (i.e., mass of tephra fallout) is 1.1-2.3 times of the estimated volume change in the SMSZ. As discussed by Rivalta & Segall (2008), R_v is always greater than 1. However, in the original manuscript, we have shown that the estimated volume change in the SMSZ is 6-8 times of the DRE of tephra fallout, or equivalently $R_v \sim 0.12-0.14 < 1$. Therefore, we think the volume difference is not a result of magma compressibility.

We again refer to the discussion in the revised manuscript (nP9L11-L24).

Page 11:

Line4: This sentence is filled with either assumptions or claims that aren't yet supported in the manuscript such as the existence of a "crystal-rich mush and crystal-poor pool"

We thank the reviewer for the comment.

As noted earlier in the rebuttal letter, we have revisited the magma plumbing system recently summarised by Kawaguchi et al. (2021) and slightly modified the interpretation in the revised manuscript (nP6L13-L18). Specifically, we noted the process of magma mixing for post-caldera volcanisms beneath Aso (Miyoshi et al. 2011; Miyoshi et al., 2012; Kawaguchi et al., 2021) and inferred discrete magma transport between a chamber with volatile-poor silicic magma and a storage zone of mixed-magma (i.e., SMSZ).

On the other hand, we note that the crystal-rich mush domain likely remains at a deeper depth (i.e., >10 km) after the most recent Aso-4 caldera forming eruption (Ishibashi et al., 2018). This is briefly noted in the revised manuscript (nP6L13-L19). We also append a short paragraph in the introduction to lay out the background on the transition of magma plumbing system from caldera-forming eruptions to post-caldera volcanisms (nP2L7-L13).

Line 6: Is this claim discussed further somewhere? If so, I missed it: "The duration of each deformation event (~ 50 s) is much longer than what is expected for crustal earthquakes of similar size and such a slow deformation"

We thank the reviewer for this comment.

To put the discussion in a proper context, we have rearranged the paragraph in the revised manuscript (nP7L24-L28).

The perspective could be improved with adding a paragraph or two about how the lessons here could be applied to specific other volcanoes (if possible). Applying the techniques to other volcanoes is mentioned, but what other volcanoes have a similar eruptive style and might be the best targets to investigate? Also, there are many types of eruptions or plumbing systems for which these types of analysis would not work and should be mentioned as well. If additional space is needed, I suggest dropped Figs. 5 and 6 below.

We thank the reviewer's comment.

We have added a paragraph to discuss the applicability of the detection and identified volcanoes where the inference may be possible (nP12L17-L24). Following the suggestion by the review 2 and reviewer 3, we have removed Fig. 6 in the original manuscript.

The data availability statement is not clear: Are the tilt and seismic waveforms available from the link provided? Also the statement that data products are available by request is no longer considered a best practice (for example, it is not allowed by AGU). These data products are not required to be made available in a public repository, but it would add great value if they were. Does UCL have such a repository?

We thank the reviewer for the comment.

The data availability and code availability have been restructured according to Nature Communications requirement. The broadband and tilt waveforms are available from the provided link. The catalogue and code can be obtained upon request.

Fig.1: What are BCU and BYA chambers? They are not mentioned in the caption. Also, what is the depth of the low velocity zone, and other features listed in the legend (maybe refer to Fig. 3)?

We thank the reviewer for this comment.

We refer the depths info to Fig. 3c.

Fig. 2: Some more details are needed in the caption. How many events are stacked together here?

We thank the reviewer for the comment.

We have revised the caption. The number of events used for the inflation and deflation stacks is 671 and 967, respectively.

How were events horizontally aligned?

We clarified in the caption that the traces are aligned with respect to the onset of LP.

Fig. 3: Where do the horizontal and vertical displacements in a and b come from? GNSS? Or integrated from seismometer?

We thank the reviewer for this comment.

The horizontal and vertical displacement are obtained from the broadband seismograms and the data processing has been detailed in the Method section (Estimate the static displacement offset from broadband seismograms).

I could be helpful in d to show the depths of the features shown in a, b, and c: where are the low velocity zone, BYA and BCA chambers, inverted source location (Red Cross) and new Mogi (black circle)?

We thank the reviewer's comment.

We show the depths of all features in Fig. 3c with a legend.

Why is there an aquifer labeled in d? I don't think the aquifer is mentioned in the caption or the text.

The aquifer near the crack-like conduit has been widely discussed in the literature. We have noted the aquifer in the revised manuscript (nP6L6-L12)

I'm also confused about what is happening in e, f, and g. What are the red and blue dotted lines at the line labeled LPT? What are the arrow at the SPT line? What are the arrows in-between the LPT and SPT lines? What physical processes do these arrows/features represent?

We thank the reviewer's comment.

We have modified the caption and append the details (Fig 4 in the revised manuscript).

Fig. 4a: where does the accumulative net volume change come from? The data used to calculate this should be mentioned in the caption.

We thank the reviewer for the comment.

As already noted in the original manuscript (P8L8-L12), the accumulative net volume change is calculated from the monthly volume change associated with the monthly inflation/deflation event stacks. We also mentioned this information in the revised manuscript (nP9L1-L5).

I do not user stand the labels that say, for example, $\Delta V_{\text{magma}} \ll \Delta V_{\text{gas}}$ during time period 1. It looks like the volume change is basically flat during this time period, so shouldn't these two volume changes be in approximate balance instead of orders of magnitude different (as implied by using \ll)?

We thank the reviewer for this suggestion.

The volume change during episode 1 is not flat. As noted in earlier reply, we have modified the text to clarify the discussion in the revised manuscript (nP9L11-L17). In particular, we have appended the net volume change from events with a high signal-to-noise ratio (i.e., subset I+II) (Fig. 5a in the revised manuscript) to facilitate the discussion.

Fig. 4b: What is "outgassing potential"? I haven't heard this term before and the phrase used in the caption is still confusing: "the moment ratio between pressurization and 10 depressurization LPTs"

We thank the reviewer for the comment.

To avoid confusion and focus on the observations, we replace Fig. 5b with crater bottom (wall) temperature and append discussions in the revised manuscript (nP8L14-L18).

Fig. 4d: Where is the volume change rate being measured? SMSZ? The caption says the black crosses are the "inflation event" but I think this should be "inflation events."

We thank the reviewer for this comment.

Yes, the volume change rates refer to the SMSZ. We have used circle and cross to indicate the averaged rate and single-event rate, respectively. Fig. 4d has now been rearranged as Fig. 6b in the revised manuscript.

"What is the geodetic data used to estimate the green cross? Is this the same tilt data described in this paper or something else, like GNSS? What is the time period of the geodetic data?"

The geodetic data corresponding to the green cross is levelling data. The time period of the geodetic data is from 1958 to 2004. This has been clarified in the revised manuscript (Fig. 6b).

In general, it should be noted that these volume change rates are being measured over vastly different time periods and the time periods should be mentioned in the caption.

We thank the comment by the reviewer.

We have modified the figure and the caption to highlight the difference. In addition, we use a different symbol to highlight the volume change rate for a single event (i.e., time scale of ~ 100 s).

Fig. 4e: Considering there are only 2 data points, it does not seem wise to draw a line between them.

The line is not a fitting line. It simply indicates a 1-to-1 relationship between the estimated mass change in SMSZ and the mass of tephra fallout. If the estimated mass change in SMSZ is equal to the mass of tephra fallout, the data point will fall on the line.

Figures 5 and 6 have minimal discussion in the text and do not seem to factor into the key conclusions mentioned in the abstract -- could they be removed? They probably deserve further discussion in a separate paper. Fig. 5 has a huge amount of information that isn't discussed in the main text.

We thank the review for the suggestion.

We have removed Fig. 6 in the original manuscript. On the other hand, we have expanded the discussions for Fig. 5 (now Fig. 7) in the revised manuscript (nP11L26-nP12L5).

In particular, I think Fig. 6 is confusing and possibly misleading. It seems to take a single volcanic system and wildly extrapolate it to all systems worldwide. This figure seems to imply that the magma plumbing system of Aso is relevant to all types of eruptions from Rhyolites to flood basalts which is clearly not true -- we have enough information to know that the plumbing of Aso is not widely applicable to all volcanoes. The authors should consider what is the point this figure is trying to make and if it is already made successfully in the text. If the point is that "Composition, viscosity, rheology and tectonic settings govern the recurrences of episodic deformation" then this point can be made adequately in the text without confusing the reader.

Following the suggestion by the reviewer 3 and another reviewer, we remove Fig. 6 in the original manuscript since the main point has been discussed in the text. Instead, we add a sentence in the revised manuscript to emphasize that magma composition, viscosity, rheology and tectonic settings govern the recurrences of episodic deformation (nP10L12-L14).

Fig. 6: "providing the glue" is a confusing phrase here

Please see above reply and Fig. 6 in the original manuscript has been removed.

Reviewer #3 (Remarks to the Author):

*Review comments on Episodic transport of discrete magma batches beneath Aso volcano
By Jieming Niu, Teh-Ru Alex Song*

This manuscript is quite interesting and is sufficiently valuable to publish on Nature Communications.

Major comments

1) Source of tilt offset is discussed in the relation of crystal rich and crystal poor zones. I cannot well understand how to relate the source of tilt off set with the rate of crystal. The reference 48 investigate volatile from the viewpoint of petrology. If source of tilt offset is related with volatile-rich and volatile-poor zones, this might be better understood.

We thank the reviewer for the comments. As pointed out by the reviewer, the reference 48 (Kawaguchi et al., 2021) did not directly address crystal-rich or crystal-poor zone in the magmatic system beneath Aso caldera.

As noted earlier in the rebuttal letter, following latest petrological studies (e.g., Kawaguchi et al., 2021), we note magma mixing between volatile-poor silicic magma at the magma chamber (~ 4-10 km) and volatile-rich basaltic magma coming from a deeper reservoir (> 10 km). Furthermore, the storage depth of mixed-magma is determined at ~2-4 km depth (or ~ 1-3 km BSL). The source of tilt is interpreted as a transport of magma batch between the top of magma chamber and the storage zone of mixed magma (SMSZ discussed in the manuscript). We have modified the interpretation in the revised manuscript (nP6L13-L18, nP6L23-L28). The transport process is also elaborated in the revised manuscript (nP7L3-L28).

2) Source of tilt offset is also discussed on brittle-ductile transition zone. However, the authors assumes that tilt offset is induced by volume change of a combination model of tensile crack and explosive source in elastic medium and this model does not include fracture. Brittle and ductile are manners of fracture.

We thank the comment by the reviewer.

Here we disagree with the reviewer's comment. As noted by Aki & Richard (2002, Quantitative Seismology, chapter 3) and many other seismology textbooks, seismic moment tensor is a general force-equivalent representation of internal sources, including fracture, explosion and tensile-crack inside the earth. On the other hand, to reiterate, our seismic moment tensor inversion shows that the source of the tilt-offset has a predominant volumetric component and (~80%) and a minor normal-fault component (~20%).

Regarding to the model of elastic deformation, source of tilt offset should be discussed on difference in elastic constants along the magma plumbing system.

As noted by Aki & Richard (2002, Quantitative Seismology, chapter 3), the elastic moduli used in the force-equivalent representation of internal sources are constants appropriate for the wall rock (or unaltered rock). While it is likely that the elastic constant within the magma plumbing system may vary, it does not change seismic wave excitation in the frequency band of our interest (i.e., > 50 sec). We have briefly discussed the effect of elastic constant on source inversion and the volume change in the revised manuscript (nP5L26-L29).

The inverted source location and geometry only depends on the displacement or tilt ratio among different stations or/and different components. We also inverted the source with a different velocity structure and the effect of elastic constant on the solution is minimal.

3) *The significance of comparison of 2011-2016 eruptivity of Aso with basaltic eruptivity is not well understood. As mentioned in the text, long-term eruption rate of andesitic volcanoes is lower than basaltic volcanoes. I cannot find a significance of comparison of eruption rate between andesitic and basaltic volcanoes in this manuscript.*

We thank the reviewer for the comment.

We have added a paragraph and noted the relevance of the comparison in the revised manuscript (nP11L26-nP12L5). Comparing mass discharge rate between basaltic and basalt-andesitic volcanoes help elucidate the regime where the mass discharge rate can be approximated by the mass flow rate.

The eruption 2011-2016 is an eruptive activity of Aso, however it does not cover all the eruptivity of Aso. If compared, the eruption 2011-2016 should be compared with past eruptivity of Aso or long-term eruptivity of the volcano.

We largely agree with the reviewer's comment. In Fig. 4d in the original manuscript, we did compare the averaged eruption output rate in 2011-2016 against the output rate in historical eruptions and long-term eruption output rate over the geological time scale. These comparisons have been noted in P8L8-L18 in the original manuscript. In the revised manuscript, we also appended the magma discharge rates of the 1979 and 1989 eruptions in the Fig. 6 for comparisons. We have modified the text accordingly (nP11L16-L20).

Minor comments

P3L20

"natural period" -> band width

The use of natural period was correct. It is used to describe the resonant period of a seismometer (pp.175, Lay and Wallace, 1995).

P4L12

LP signal east-west is much weaker, but north-south is stronger than N.ASIV.

We thank the reviewer for the comment. The change has been made in the revised manuscript (nP4L17-L21).

P4L13 "These observations strongly indicate that the source of the tilt offset is spatially separated from the LPT source"

It is possible, but is it necessary to examine source difference between tilt offset and LPT?

We thank the reviewer for the comment.

As discussed in P4L8-L13 in the original manuscript, the difference in the amplitude between LPT and tilt offset among different seismic station or/and channels is a direct indication that the source of tilt offset must differ from the source of LPT, either in location or mechanism. We have rephrased the sentence to clarify this conjecture (nP4L23-L25).

P4L14 "which is near the active Naka-dake first crater"

Show references or see Method.

We thank the reviewer's comment and have cited relevant references in the revised manuscript (nP4L25).

P5L3 global waveform stacks

What do you mean "global"? How many LPTs were stacked?

We thank the reviewer for the comment. The term "global" is used to highlight waveform stacks over the period of unrest (2011-2014 August) and Strombolian eruption (Nov. 2014-

Apr 2015). The number of LPT events used for stacking is shown in Fig. 2 and noted in the caption.

P5L9 relatively steady

Almost the same?

We thank the reviewer's comment. We have rephrased the sentence in the revised manuscript (nP5L4).

P5L21 Fig. S4

Fig. S4 is "Synthetic amplitude-distance decay against static and filtered waveforms". Inserting Fig. S4 explains why you choose 100-200s ULP band?

We thank the reviewer for the comment.

We have modified the sentences accordingly in the revised manuscript (nP4L3-L6).

P7L2 "SMSZ"

"SMSZ" firstly appeared here. This should be explained.

We thank the reviewer for the comment. The SMSZ is first defined and noted in the revised manuscript (nP6L21).

P7L5 intense

Is this word necessary? What do you mean "intense unrest"?

We thank the reviewer for the comment.

We have added a brief note to define specific activities in the revised manuscript (nP8L2-L4).

P7L18 "a deeper reservoir (i.e., a crystal-rich mush) and temporarily stalled in the SMSZ (i.e., a crystal-poor pool)"

The reference 48 investigate volatile. How does it relate to crystal? Are there any evidence for crystal-poor in the SMSZ?

We thank the reviewer for the comment.

As noted earlier in the rebuttal letter, we have modified the interpretation and the discussions in the revised manuscript (nP6L13-L19).

P8L1-L7

For estimation of volume increase, Method should be referred.

We thank the reviewer for the comment. This statement has been referred to the Method in the revised manuscript (nP9L1-L5).

P8L13-L22

This hypothesis is true. As the authors show clearly, LPT is a short-lived phenomenon. What does the author mention, comparing short-lived volume change with long-term eruptivity?

We thank the reviewer for the comment.

As noted in the original manuscript LPT (termed VLP in the revised manuscript) has been observed over multiple eruption cycles since 1920s (Sassa, 1935; Sakaguchi et al., 2008). This provides a means to detect synchronous VLP and inflation/deflation event and evaluate single-event volume change for historical eruptions. This point has been addressed in the revised manuscript (nP12L11-L16).

P10L11-L13 "While the estimated mass flow rate in Aso is lower than those estimated in basaltic eruptions by an order of magnitude (Fig. 5), such a difference is consistent with the disparity in the average volcanic output rate between basaltic eruptions and andesitic eruptions"

In this case, Aso means activity 2011-2016? Which eruption does “basaltic eruption” indicate?

We thank the reviewer for the comment.

Yes, the mass discharge rate was referred to Aso activity 2011-2016. In the original manuscript, the basaltic eruptions refer to data shown in Fig 5 (solid diamonds in Fig. 7 in the revised manuscript) where both the decompression rate and the mass discharge rate are available (Barth et al., 2019). On the other hand, the relevant discussion has been revised in the revised manuscript (nP11L26-nP12L5).

The first half is comparison of short term activity. Second half is long-term comparison. Long-term comparison may be true. But short-term comparison is case-by-case. It is not necessary to be consistent.

As noted above in the rebuttal letter, the relevant discussion can be referred to the revised manuscript (nP11L26-nP12L5).

P22L11

Takuhiro -> Takahiro

We thank the reviewer for the comment.

This has been modified accordingly.

Figures

P24L9 October 8, 2016

Need time of onset of the eruption.

The onset of the eruption is marked by the red line. We have modified the caption of Fig. 1b accordingly.

(b) N.ASHV.LE is tilt of east-side down at station N.ASHV?

Yes, we have also clarified this in the caption.

Is Figure 6 needed?

We thank the reviewer for the comment.

As suggested by the reviewers, this figure has been removed.

REVIEWERS' COMMENTS

Reviewer #2 (Remarks to the Author):

I thank the authors for addressing my comments in the first round of review. I think they have adequately modified their manuscript and I recommend publication in its present form.

Reviewer #3 (Remarks to the Author):

I think the manuscript is well revised according to the comments by the reviewers. This manuscript documents magma plumbing system of Aso volcano from the deep chamber to the shallow conduit and reveals SMSZ to connect the two magmas. The two stages; pre-eruptive and Strombolian eruption are well separated by extracting minor offset of the tilt and seismic events. This attains at a level to publish on nature communications, supporting high-precision data.

The followings may be mistyped.

P2 L27 sthe->the

Fig1 L6 NewMogi -> New Mogi

Fig2 L8 acasual -> casual

In the following response, the comments by the reviewers are shown in *italic*, our responses are shown in **bold**. The page and line numbers in the original manuscript are noted as **P?L??**. The responses discussed in the rebuttal letter are also highlighted in yellow in the revised manuscript.

Reviewer #2 (Remarks to the Author):

I thank the authors for addressing my comments in the first round of review. I think they have adequately modified their manuscript and I recommend publication in its present form.

We thank the reviewer for the positive comments and support for the publication.

Reviewer #3 (Remarks to the Author):

I think the manuscript is well revised according to the comments by the reviewers. This manuscript documents magma plumbing system of Aso volcano from the deep chamber to the shallow conduit and reveals SMSZ to connect the two magmas. The two stages; pre-eruptive and Strombolian eruption are well separated by extracting minor offset of the tilt and seismic events. This attains at a level to publish on nature communications, supporting high-precision data.

We thank the reviewer for the positive comments and support for the publication.

The followings may be mistyped.

P2 L27 sthe->the

This has been corrected as P2L27 in the revised manuscript.

Fig1 L6 NewMogi -> New Mogi

This has been corrected as P30L8 in the revised manuscript. “NewMogi” in the figure has also been replaced by “New Mogi”.

Fig2 L8 acasual -> casual

This has been corrected as P31L8 in the revised manuscript.

Note we slightly revise Fig. 5b and append the temperature data from ground-based thermal camera (Cigolini et al., 2018), filling the data gap between late 2014 and 2016. We append two references summarizing latest efforts on deep low-frequency earthquakes (P12L25). We slightly refine and make a more precise description of source mechanism (P6L3-5) and append uncertainty estimates in the supplementary Table 2.

Finally, we have also followed the checklist and slightly reformat the manuscript. As suggested by the editorial team, we include an image to be featured in Nature Communications. We believe the image vividly illustrates the background crater-lake and diverse eruption styles in Aso volcano. The original images are credited to Dr. Akihiko Yokoo in Aso Volcano Observatory.